# QUERY-KONTEXT: AN UNIFIED MULTIMODAL MODEL FOR IMAGE GENERATION AND EDITING

## ABSTRACT

Unified Multimodal Models (UMMs) have demonstrated remarkable performance in text-to-image generation (T2I) and editing (TI2I), whether instantiated as assembled unified frameworks which couple powerful vision-language model (VLM) with diffusion-based generator, or as naive Unified Multimodal Models with an early fusion of understanding and generation modalities. We contend that in current unified frameworks, the crucial capability of multimodal generative reasoning which encompasses instruction understanding, grounding, and image referring for identity preservation and faithful reconstruction, is intrinsically entangled with high-fidelity synthesis. In this work, we introduce Query-Kontext, a novel approach that bridges the VLM and diffusion model via a multimodal *"kontext"* composed of semantic cues and coarse-grained image conditions encoded from multimodal inputs. This design delegates the complex ability of multimodal generative reasoning to powerful VLM while reserving diffusion model's role for high-quality visual synthesis. To achieve this, we propose a three-stage progressive training strategy. First, we connect the VLM to a lightweight diffusion head via multimodal kontext tokens to unleash the VLM's generative reasoning ability. Second, we scale this head to a large, pre-trained diffusion model to enhance visual detail and realism. Finally, We introduce a low-level image encoder to improve image fidelity and perform instruction tuning on downstream tasks. Furthermore, we build a comprehensive data pipeline integrating real, synthetic, and curated open-source datasets, covering diverse multimodal reference-to-image scenarios, including image generation, instruction-driven editing, customized generation, and multi-subject composition. Experiments show that our approach matches strong unified baselines and even outperforms task-specific state-of-the-art methods in several cases.

## 1 INTRODUCTION

Unified Multimodal Models (UMMs) have recently achieved notable progress in both image generation (T2I) Podell et al. (2024); Esser et al. (2024a); Chen et al. (2024a); Ho et al. (2022); BlackForest (2024); Deng et al. (2025); Gao et al. (2025); Liu et al. (2025b); Pan et al. (2025); OpenAI (2025a); Wu et al. (2025a) and editing (TI2I) Brooks et al. (2023); Zhang et al. (2023); Ye et al. (2025); Liu et al. (2025a); Labs et al. (2025); DeepMind (2025); Kuprashevich et al. (2025a); Wang et al. (2025a); Wei et al. (2024). Two prominent design paradigms have emerged from this work. The first assembled unified framework leverages external diffusion transformers, such as MMDiT Esser et al. (2024a); William & Xie (2023), which are paired with off-the-shelf vision–language models (VLMs) or large language models (LLMs) to provide semantic conditioning. The second paradigm, naive UMMs, integrate generation and understanding more tightly through mixed-modal early-fusion transformers Zhou et al. (2024); Deng et al. (2025); Team (2024); Chen et al. (2025e); Wang et al. (2024e); Tong et al. (2024); Ma et al. (2024b), where autoregressive modules with strong reasoning ability are jointly trained with diffusion modules specialized in visual synthesis.

While these paradigms expand task coverage and streamline deployment, they also entangle multimodal generative reasoning and high-fidelity rendering. Consequently, the unique strengths of VLMs (semantic understanding, grounding, structured reasoning Wang et al. (2024b); Bai et al. (2023; 2025); Chen et al. (2024d;e); Yao et al. (2024b); Zhao et al. (2024a); Yao et al. (2024a)) and diffusion models (photorealistic synthesis and detail fidelity William & Xie (2023); Chen et al. (2023a); Nichol

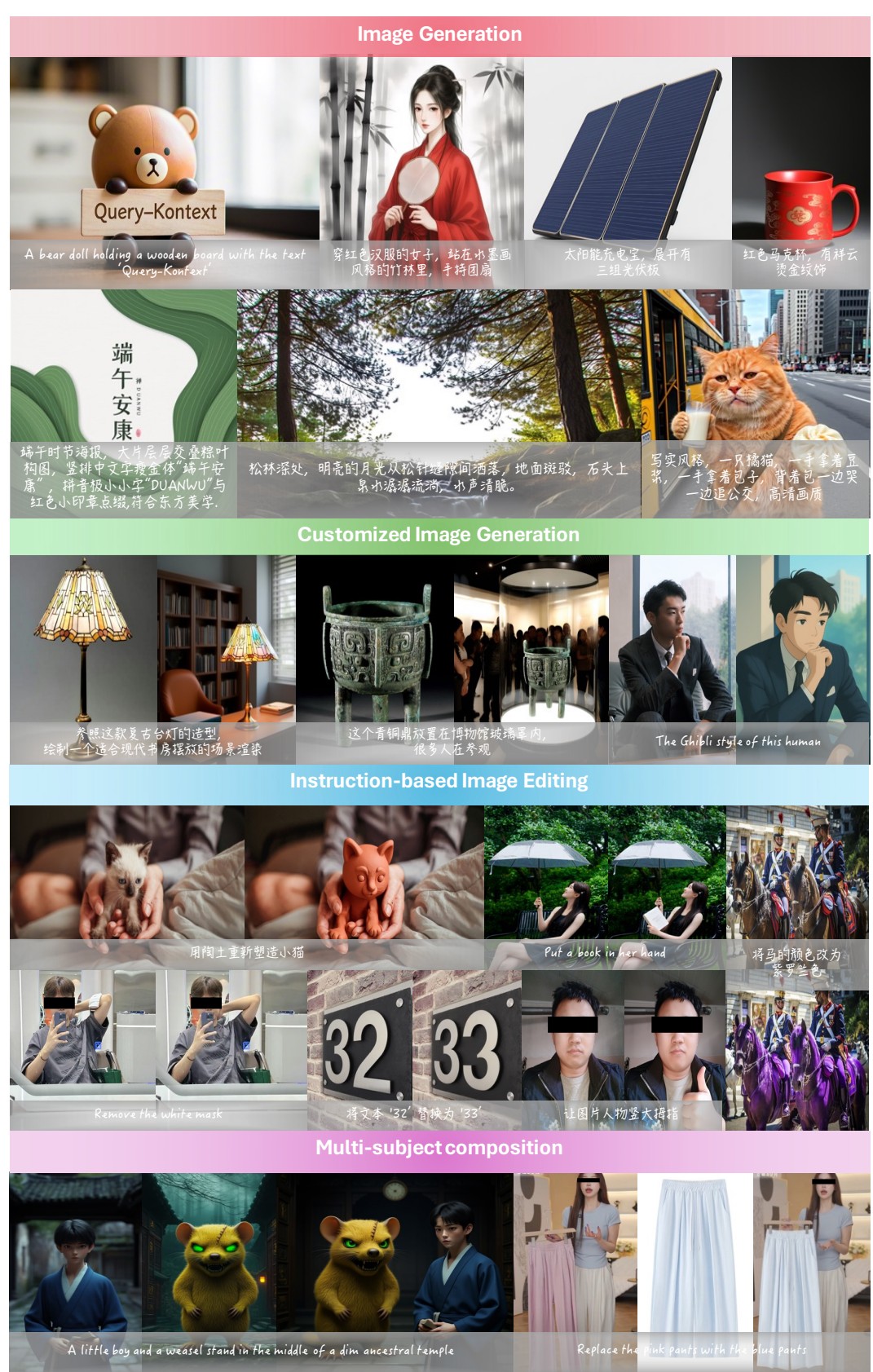

Figure 1: Showcase of Query-Kontext model on multimodal reference-to-image tasks.

& Dhariwal (2021); Ho et al. (2020); Chen et al. (2024c;b); Ho et al. (2022)) cannot be fully exploited. We identify two sources of this limitation. First, *assembled* unified frameworks typically use a frozen VLM or LLM as a static feature extractor, narrowing the conditioning signal to only high-level semantics for the diffusion generator. Second, *native* UMMs force generative reasoning and visual rendering to be optimized jointly, introducing capacity competition and hindering generalization, particularly when tasks demand both fine-grained edits and strong semantic control. While attempts to mitigate these issues through methods like mixture-of-experts (e.g., LlamaFusion Shi et al. (2024)) or mixture-of-transformers (e.g., BAGEL Deng et al. (2025)) have been made, they only partially alleviate the tension.

In this work, we propose **Query-Kontext**, an economic ensemble UMM that leverages the multimodal *"kontext"* composed of semantic and coarse image conditions to cleanly decouple the generative reasoning of VLM from the high-fidelity rendering of diffusion model. To realize this separation, we develop a three-stage progressive training strategy. **Stage 1:** Bridge the VLM to a lightweight diffusion head through *"kontext"* tokens. Using parameter-efficient fine-tuning (LoRA) Hu et al. (2021), we **unleash** the potential of VLM and steer it toward multimodal generative reasoning skills such as instruction following, spatial grounding, and identity-preserving image referencing. **Stage 2:** **Scale** the lightweight head to a well-trained large diffusion model (roughly $10\times$ more parameters). We re-align both the text and *"kontext"* tokens from the VLM to the scaled diffusion model by using text-to-image generation and image-reconstruction objectives. **Stage 3:** Introduce a low-level image encoder AI (2024) that injects fine-grained structural and textural cues into the diffusion model while keeping the VLM frozen. This step strengthens identity preservation Ye et al. (2023); Wang et al. (2024d); Wu et al. (2025c); Song et al. (2025) and reconstruction fidelity in Liu et al. (2025a); Ye et al. (2025); Huang et al. (2024b); Xu et al. (2025) challenging editing scenarios.

In summary, our contributions are:

- We propose **Query-Kontext**, an economic ensemble UMM that decouples multimodal generative reasoning in VLMs from the high-fidelity visual rendering performed by diffusion models.

- We present a three-stage progressive training strategy that progressively aligns the VLM with increasingly capable diffusion generators while amplifying their respective strengths in generative reasoning and visual synthesis.

- We present a deliberate dataset curation scheme to collect real, synthetic, and carefully filtered open-source datasets to cover diverse multimodal reference-to-image scenarios.

## 2 DATA CURATION

We constructed a multimodal reference-to-image dataset (as summarized in Table 1) comprising a mixture of real, synthetic, and carefully curated open-source datasets. This dataset spans five categories of tasks: text-to-image generation, image transformation, instruction editing, customized generation, and multi-subject composition.

**Text-to-Image Generation and Image Reconstruction.** We collected 30M open-source English image-text pairs (including ShareGPT-4o-Image (Chen et al., 2025b), BLIP-3o (Chen et al., 2025a), among others) as well as 170M in-house Chinese image-text pairs for text-to-image generation and image reconstruction tasks. The in-house data underwent extensive quality filtering based on image resolution, clarity, aesthetic score, watermark detection, and safety compliance. Among these Chinese data, 150M belong to general categories (balanced across diverse domains), and 20M come from specific vertical domains (e.g., artistic styles, logos, automobiles, text-containing images, celebrities, posters, etc.).

**Image Transformation.** Following the MetaQuery (Pan et al., 2025), we constructed naturally occurring image pairs from web corpora (Chen et al., 2025c; Li et al., 2024a) and generated corresponding open-ended transformation instructions by leveraging multi-modal large language models (MLLMs). Specifically, we clustered images that share the same accompanying caption from sources like MMC4-core(Chen et al., 2025c), OmniCorpus-CC (Li et al., 2024a) and OmniCorpus-CW (Li et al., 2024a) by using SigLIP (Tschannen et al., 2025) image features, then filtered these clusters by a similarity threshold to obtain 0.8M image transformation triplets (*see Appendix for example triplets B.1*).

**Instruction Editing.** For the image editing instruction task, we first aggregated approximately 3M image-instruction-image triplets from open-source datasets, including NHR-Edit (Kuprashevich et al., 2025a)(358k samples), GPT-Image-Edit (OpenAI, 2025b)(1.5M samples), MagicBrush (Kawar et al., 2023)(10k samples), and OmniEdit (Wei et al., 2024)(1.2M samples). We further filtered the MagicBrush subset using CLIP-based image and text similarity scores, and translated all datasets' instructions into Chinese using a large language model. Building upon the methodologies of (Kuprashevich et al., 2025a; Wei et al., 2024; Liu et al., 2025a), we then constructed a synthetic data pipeline tailored for native Chinese instruction editing, producing an additional 300k high-quality triplets (*see Appendix B.2 for details on the data pipeline*). Finally, inspired by UniReal (Chen et al., 2025d), we extended our dataset with video-based clusters derived from raw videos to cover more non-rigid editing tasks (e.g., motion changes, viewpoint shifts, view transitions such as zoom-in and zoom-out (*see Appendix B.2 for the video data examples*).

**Customized Generation.** We leveraged the open-source Subject-200K (Tan et al., 2024) and UNO-1M (Wu et al., 2025c) datasets for customized (subject-driven) image generation. In addition, we augmented our data with portrait reference triplets synthesized using a dedicated model [1], which generates reference images of specific individuals. Through this approach, we accumulated approximately 0.3M portrait reference samples that maintain high facial similarity to the source subjects while exhibiting substantial diversity in poses, attire, and other attributes.

**Multi-Subject Composition.** Finally, we addressed multi-subject image composition using the open-source MUSAR-Gen (Guo et al., 2025) dataset and a new synthetic data pipeline (*see details in Appendix B.3*). This yielded 40k multi-subject reference examples, each featuring compositions of multiple humans, objects, and complex scenes, thereby enriching the dataset's coverage of realistic multi-entity interactions.

Table 1: **The data outline about each training stage.** [†] denotes only the Chinese prompt.

| Stage | Task | Data source | Size |
|---|---|---|---|
| 1, 2, 3 | Image generation, Image reconstruction | ShareGPT-4o-Image, BLIP3o | 30M |
| | | in-house real data[†] | 170M |
| 1 | Image transformation | mmc4, OmniCorpus | 800K |
| 3 | Instruction editing | NHR-Edit, GPT-Edit, OmniEdit | 3M |
| | | In-house video data | 2M |
| | | In-house real data | 300K |
| | Customized generation | subject200k | 200K |
| | | in-house real data | 1.8M |
| | Multi-subject composition | MUSAR-Gen | 29K |
| | | GPT-4o synthesized data | 40K |

## 3 QUERY-KONTEXT

In this work, we propose Query-Kontext, a unified multimodal model for image generation and editing that delegates multimodal generative reasoning to the VLM while reserving the diffusion model's capability for high-quality visual synthesis. In Sec 3.1, we present the architectural design of the Query-Kontext model (Figure 2). In Sec 3.2, we design a three-stage progressive learning strategy and introduce the details of training recipe (Figure 3). In Sec 3.3, we introduce the implementation details of model hyper-parameters and infrastructures.

### 3.1 ARCHITECTURE

As shown in Figure 2, Query-Kontext comprises four main components: a Multimodal Large Language Model (MLLM), a connector module, a Multimodal Diffusion Transformer (MMDiT), and a low-level image encoder (VAE). The MLLM is initialized with the Qwen2.5-VL model Bai et al. (2025), which encodes and fuses multimodal inputs including the text prompt, input image(s), and a set of learnable query tokens. The output is a fixed-length sequence of *kontext* tokens $Q = \{q_1, \ldots, q_K\}$ which serves as coarse image-level conditioning for the diffusion decoder while

---

[1] https://console.bce.baidu.com/qianfan/modelcenter/model/buildIn/detail/am-t3uhhjzbys6w

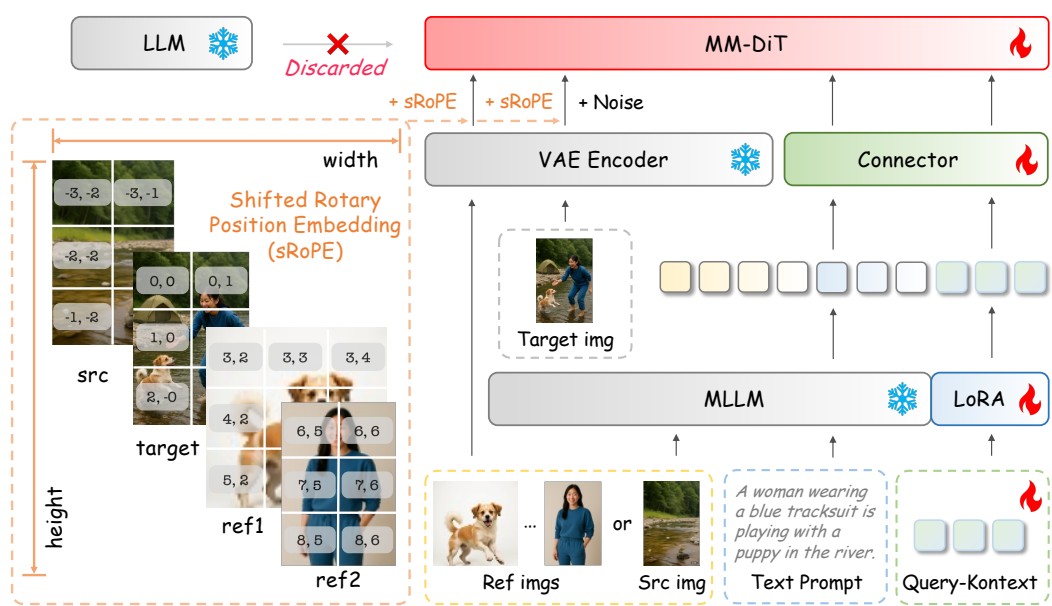

Figure 2: **The overall framework of the unified multi-modal to image generation and editing model, Query-Kontext.**

providing high-level semantic cues. Intuitively, the $kontext$ tokens $Q$ encode what content should appear in the output image (the semantic information from the text prompt) and how the output should incorporate visual cues from the provided input images, as enforced by the training supervision in Sec. 3.2. The $kontext$ $Q$ and text tokens $T$ are passed through a lightweight connector module to align them with the diffusion model's latent space. In practice, we concatenate the noisy latent, the low-level image latent with sRoPE, the text latent $T$ and the "kontext" token $Q$ from the connector, and subsequently feed them into MMDiT, thereby enriching the semantic context and image details, which are made available for the diffusion model. We initialize the diffusion model using our in-house MMDiT model and replace its original text encoder with the MLLM (*training details for this alignment are discussed in Sec. 3.2*). We concatenate the sequence of text $T$ and $kontext$ tokens $Q$ from the MLLM with: (i) the noisy image latent at the current diffusion step $t$, and (ii) the low-level visual feature tokens extracted from the input image(s) by the VAE. The concatenated sequence is then fed into the MMDiT model in an in-context manner Labs et al. (2025); Zhang et al. (2025), allowing the diffusion model to attend to both the textual prompt and the visual cues from the input images.

Moreover, we distinguish between naive UMMs and assembled UMMs in Table 2. The comparison highlights whether each model trains from scratch, freezes parameters, or adapts pretrained components, and specifies the flow of information through text embeddings (TE), low-level image embeddings (LE), and query embeddings (QE). In particular, query embeddings naturally unleash the in-context learning capabilities of the VLM, enabling the model to reason over multimodal inputs and generate coherent images. Unlike prior methods, **our Query-Kontext integrates query embeddings alongside text and low-level image embeddings, while effectively decoupling understanding and generation modules for improved efficiency and flexibility.**

Furthermore, we design a *shifted 2D Rotary Position Embedding* (RoPE) scheme Su et al. (2024); Wu et al. (2025c) to incorporate multi-image positional conditioning and avoid confusion among multiple reference images (as illustrated in Figure 2). In the standard diffusion architecture, each spatial position of a latent feature map (with size of $h \times w$) is identified by a 2D index $(i, j)$, where $i \in [0, w-1]$ and $j \in [0, h-1]$. We introduce a task-specific prior to adjust these coordinates based on the fidelity requirements of the input images. For tasks requiring pixel-level fidelity to an input image (e.g., instruction-based editing), we treat the input image as a *source image*, denoted $img_{src}$. For tasks requiring identity preservation (e.g., personalized generation or multi-image composition), we treat the input image as a *reference image*, denoted $img_{ref}$. We then shift the coordinate indices

Table 2: **Comparison of mainstream unified multimodal models on the modeling paradigms and the information flow.** 🔥 denotes training from scratch, ❄ indicates freezing the parameters during training and ❄ → 🔥 represents training from a pretrained model. For the input modalities, "TE" refers to text embeddings, "LE" to low-level image embeddings, and "QE" to query embeddings.

| Method | Module | | | Information | | |
|---|---|---|---|---|---|---|
| | **Understanding** | **Connector** | **Generation** | **TE** | **LE** | **QE** |
| *Native UMMs* | | | | | | |
| Janus-Pro Chen et al. (2025e) | 🔥 | - | 🔥 | ✓ | ✗ | ✗ |
| OmniGen2 Wu et al. (2025b) | ❄ → 🔥 | - | 🔥 | ✓ | ✓ | ✗ |
| BAGEL Deng et al. (2025) | 🔥 | - | 🔥 | ✓ | ✗ | ✗ |
| *Assembled UMMs* | | | | | | |
| Metaquery Pan et al. (2025) | ❄ | 🔥 | ❄ → 🔥 | ✓ | ✗ | ✓ |
| Step1X-Edit Liu et al. (2025a) | ❄ | 🔥 | ❄ → 🔥 | ✓ | ✓ | ✗ |
| Uniworld-v1 Lin et al. (2025) | ❄ | 🔥 | ❄ → 🔥 | ✓ | ✓ | ✗ |
| FLUX.1 Kontext Labs et al. (2025) | ❄ | - | 🔥 | ✓ | ✓ | ✗ |
| Qwen-Image Wu et al. (2025a) | ❄ | - | 🔥 | ✓ | ✓ | ✗ |
| **Query-Kontext (Ours)** | ❄ → 🔥 | 🔥 | ❄ → 🔥 | ✓ | ✓ | ✓ |

of the VAE latent for each image type accordingly: for reference image latents, we shift indices into the positive quadrant, whereas for the source image latent, we shift into the negative quadrant.

As shown in the left subfigure of Figure 2, we adopt two concatenation orders depending on task requirements: 1) for the tasks requiring strict pixel fidelity (e.g., instruction-based editing), we $\text{concat}(img_{src}, noise)$ with a shifted RoPE in the negative spatial direction. 2) for the asks emphasizing identity preservation with flexibility (e.g., customized generation), we $\text{concat}(noise, img_{ref})$ along the positive direction. we define the coordinates for the $n$-th reference latent as:

$$\left(i_{\text{ref}}^n, j_{\text{ref}}^n\right) = \left(i + w * n, j + h * n\right) \tag{1}$$

where $i \in [0, w-1], j \in [0, h-1]$ and $n \in [1, N]$. Meanwhile, for the source image latent we shift the coordinates in the negative direction:

$$\left(i_{\text{src}}', j_{\text{src}}'\right) = \left(-i, -j\right), \tag{2}$$

where $i \in [0, w-1], j \in [0, h-1]$ and $n \in [1, N]$. Finally, we add the shifted RoPE on the feature maps of the input image latent(s) and the noisy latent at their respective shifted coordinates (i.e., added element-wise to each spatial location).

### 3.2 INDIVIDUALIZED-TEACHING CURRICULUM

As shown in Figure 3, we propose a three-stage progressive training strategy that both unlocks the generative reasoning capabilities of the VLM and progressively aligns it with increasingly powerful diffusion generators. As a result, Query-Kontext, guided by multimodal $kontext$ tokens, effectively decouples the multimodal generative reasoning of the VLM from the high-fidelity visual rendering carried out by diffusion models.

**Stage 1:** We unleash the generative reasoning potential of the MLLM through two key architectural designs: we first use learnable query tokens ("kontext") to represent a mixture of semantic cues and coarse-grained image conditions, and then align the output $kontext$ tokens with a lightweight diffusion head that performs noisy prediction at a coarse level. We train all parameters of the connector, the diffusion head, and the MLLM's LoRA modules on a trio of tasks: text-to-image generation, image reconstruction, and image transformation (see Section 2). This training methodology preserves the MLLM's inherent language-vision understanding while cultivating its emergent ability for multimodal generative reasoning.

**Stage 2:** Next, we replace the lightweight diffusion head with our in-house diffusion model based on the MMDiT architecture for high-fidelity generation. In Stage 2, the full MLLM parameters (the LoRA parameters are merged into the MLLM) remain frozen, and we optimize the $kontext$ tokens, the connector, and all parameters of the large diffusion model. In preliminary experiments, we

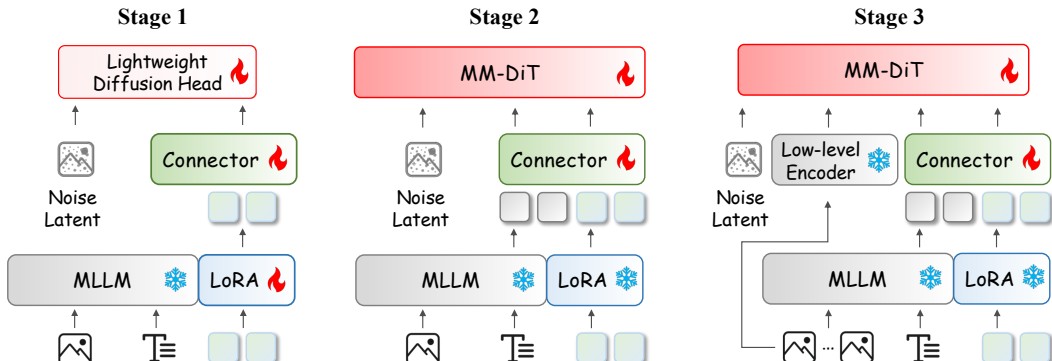

Figure 3: **Three training stages of Query-Kontext.** Note that the Diffusion Head is only used in the Stage 1. In the Stage 2 and 3, we scale up Diffusion model to 10× parameters and keep MLLM frozen to provide coarse-grained image conditions.

observed that completely freezing the diffusion model was feasible for smaller head but failed for a larger diffusion model (*the experiments details and discussion are available in Section 5*). Therefore, we allow the diffusion model to be full-parameters fine-tuning in this stage. To keep training efficient, Query-Kontext is trained only on text-to-image generation and image reconstruction tasks in this stage, which accelerates convergence and reduces training cost for fast alignment from MLLM to the diffusion model.

**Stage 3:** Finally, we introduce a dedicated low-level image encoder for source or reference images to further refine the diffusion model for high-fidelity image referring. In Stage 3, the MLLM remains fully frozen, and we optimize only the Query-Kontext tokens and the connector. Additionally, we apply the LoRA-based fine-tuning to the diffusion model itself to preserve its high-quality image synthesis ability while extending it to all our tasks. This includes not only standard text-to-image generation but also instruction-guided image editing, user-customized image generation, and multi-subject composition tasks.

### 3.3 IMPLEMENTATION

**Architecture.** We initialize the MLLM from Qwen2.5-VL-7B and implement the connector as a two-layer MLP. (*details of architecture configuration are provided in Appendix D*) Moreover, we implement the diffusion head in the stage a with a lightweight MMDiT architecture ($\sim 870M$ parameters). We set the max reference images $N = 2$ and $K = 128$ in Query-kontext $Q = \{q_1, \ldots, q_K\}$. We set rank $r_d = 256, \alpha_d = 256$ in the diffusion model's and rank $r_m = 128, \alpha_m = 256$ in the MLLM's LoRA.

**Training recipe.** The default configuration on the resolution with $512 \times 512$ is provided in Table 3. After Stage 3, we introduce a resolution upscaling stage using the same mixed multi-task dataset at a higher resolution. In this stage, the training resolution is increased to $1024 \times 1024$, the learning rate is further reduced to $1 \times 10^{-5}$, and training continues for an additional 3,000 steps with a global batch size of 256.

## 4 EXPERIMENTS

### 4.1 QUANTITATIVE RESULTS

We evaluate Query-Kontext on a comprehensive suite of benchmarks, spanning text-to-image generation, instruction-guided editing, subject-driven customization, and multi-subject composition. Specifically, we report results on GenEval, GEdit-Bench, DreamBooth, and DreamBench.

On GenEval, Query-Kontext attains an overall score of 0.88, matching the SOTA result of unified UMM (BAGEL (Deng et al., 2025)), as illustrated in Table 4. Our results are reported based on

Table 3: **The data outline and training details about each training stage.** Where, $Q.$ denotes the Query-kontext tokens, $Con.$ is Connector module.

| Stage | Stage 1 | Stage 2 | Stage 3 |
|---|---|---|---|
| Task | Image Generation Image Reconstruction Image Transformation | Image Generation Image Reconstruction | Instruction Editing Customized Generation Multi-subject |
| Type | T2I, I2I, TI2I | T2I, I2I | T2I, TI2I |
| Training Param. | MLLM's LoRA, $Con.$, Diffusion head, $Q.$ | $Con.$, MMDiT, $Q.$ | MMDiT's LoRA, $Con.$, $Q.$ |
| Global Batch Size | 512 | 1024 | 512 |
| Steps (K) | 72 | 420 | 30 |
| Learning Rate | 1e-4 | 1e-4 | 2e-5 |

Table 4: **Quantitative Evaluation results on GenEval (Ghosh et al., 2023).** † refer to the methods using the LLM rewriter.

| Model | Single Object | Two Object | Counting | Colors | Position | Attribute Binding | Overall↑ |
|---|---|---|---|---|---|---|---|
| Show-o (Xie et al., 2024) | 0.95 | 0.52 | 0.49 | 0.82 | 0.11 | 0.28 | 0.53 |
| Emu3-Gen (Wang et al., 2024f) | 0.98 | 0.71 | 0.34 | 0.81 | 0.17 | 0.21 | 0.54 |
| PixArt-$\alpha$ (Chen et al., 2024c) | 0.98 | 0.50 | 0.44 | 0.80 | 0.08 | 0.07 | 0.48 |
| SD3 Medium (Esser et al., 2024b) | 0.98 | 0.74 | 0.63 | 0.67 | 0.34 | 0.36 | 0.62 |
| FLUX.1 [Dev] (BlackForest, 2024) | 0.98 | 0.81 | 0.74 | 0.79 | 0.22 | 0.45 | 0.66 |
| SD3.5 Large (Esser et al., 2024b) | 0.98 | 0.89 | 0.73 | 0.83 | 0.34 | 0.47 | 0.71 |
| JanusFlow (Ma et al., 2025) | 0.97 | 0.59 | 0.45 | 0.83 | 0.53 | 0.42 | 0.63 |
| Lumina-Image 2.0 (Qin et al., 2025) | - | 0.87 | 0.67 | - | - | 0.62 | 0.73 |
| Janus-Pro-7B† (Chen et al., 2025f) | 0.99 | 0.89 | 0.59 | 0.90 | 0.79 | 0.66 | 0.80 |
| HiDream-I1-Full† (Cai et al., 2025) | **1.00** | 0.98 | 0.79 | 0.91 | 0.60 | 0.72 | 0.83 |
| GPT-Image† (OpenAI, 2025b) | 0.99 | 0.92 | 0.85 | 0.92 | 0.75 | 0.61 | 0.84 |
| Seedream 3.0† (Gao et al., 2025) | 0.99 | **0.96** | **0.91** | 0.93 | 0.47 | 0.80 | 0.84 |
| Qwen-Image† (Wu et al., 2025a) | 0.99 | 0.92 | 0.89 | 0.88 | 0.76 | 0.77 | 0.87 |
| BAGEL† (Deng et al., 2025) | 0.98 | 0.95 | 0.84 | **0.95** | 0.78 | 0.77 | **0.88** |
| **Query-Kontext†** | 0.98 | 0.94 | 0.81 | 0.91 | **0.85** | 0.79 | **0.88** |

Table 5: **Quantitative Evaluation results on GEdit-Bench.** G_SC is Semantic Consistency, G_PQ is Perceptual Quality, and G_O is Overall Score which is computed as the geometric mean of G_SC and G_PQ, averaged over all samples. All metrics are evaluated by GPT-4. We highlight the **best** and second-best values for each metric.

| Model | GEdit-Bench-EN (Full set)↑ | | | GEdit-Bench-CN (Full set)↑ | | |
|---|---|---|---|---|---|---|
| | G_SC | G_PQ | G_O | G_SC | G_PQ | G_O |
| Instruct-Pix2Pix (Brooks et al., 2023) | 3.58 | 5.49 | 3.68 | - | - | - |
| AnyEdit (Yu et al., 2025) | 3.18 | 5.82 | 3.21 | - | - | - |
| MagicBrush (Zhang et al., 2023) | 4.68 | 5.66 | 4.52 | - | - | - |
| UniWorld-v1 (Lin et al., 2025) | 4.93 | 7.43 | 4.85 | - | - | - |
| OmniGen (Xiao et al., 2025) | 5.96 | 5.89 | 5.06 | - | - | - |
| OmniGen2 (Wu et al., 2025b) | 7.16 | 6.77 | 6.41 | - | - | - |
| Gemini 2.0 (DeepMind, 2025) | 6.73 | 6.61 | 6.32 | 5.43 | 6.78 | 5.36 |
| BAGEL (Deng et al., 2025) | 7.36 | 6.83 | 6.52 | 7.34 | 6.85 | 6.50 |
| FLUX.1 Kontext [Pro] (Labs et al., 2025) | 7.02 | 7.60 | 6.56 | 1.11 | 7.36 | 1.23 |
| Step1X-Edit (Liu et al., 2025b) | 7.66 | 7.35 | 6.97 | 7.20 | 6.87 | 6.86 |
| GPT Image 1 [High] (OpenAI, 2025b) | 7.85 | 7.62 | 7.53 | 7.67 | 7.56 | 7.30 |
| Qwen-Image Wu et al. (2025a) | 8.00 | **7.86** | 7.56 | 7.82 | **7.79** | 7.52 |
| **Query-Kontext** | **8.36** | 7.37 | **7.66** | **8.39** | 7.35 | **7.65** |

Chinese prompts rewritten by DeepSeek [2]. On GEdit-Bench, Query-Kontext achieves the highest overall performance in instruction-guided editing, with scores of 7.66 on the English split and 7.65 on the Chinese split. These results surpass Qwen-Image (7.56 / 7.52) and GPT-Image (7.53 / 7.30), as shown in Table 5. We note that the Perceptual Quality score exhibits some shortcomings, primarily due to the lack of a reinforcement learning or supervised fine-tuning stage designed to enhance

---

[2] https://chat.deepseek.com/

Table 6: **Quantitative results for single-subject driven generation on Dreambooth.** We highlight the **best** and second-best values.

| Method | DINO ↑ | CLIP-I ↑ | CLIP-T ↑ |
|---|---|---|---|
| *Tuning-free* | | | |
| Textual Inversion Gal et al. (2022) | 0.569 | 0.780 | 0.255 |
| DreamBooth Ruiz et al. (2023) | 0.668 | 0.803 | 0.305 |
| BLIP-Diffusion Li et al. (2023a) | 0.670 | 0.805 | 0.302 |
| *Specialist Models* | | | |
| ELITE Wei et al. (2023) | 0.647 | 0.772 | 0.296 |
| Re-Imagen Chen et al. (2022) | 0.600 | 0.740 | 0.270 |
| OminiControl Tan et al. (2024) | 0.684 | 0.799 | 0.312 |
| FLUX.1 IP-Adapter BlackForest (2024) | 0.582 | 0.820 | 0.288 |
| UNO-FLUX Wu et al. (2025c) | 0.760 | 0.835 | 0.304 |
| *Generalist Models* | | | |
| OmniGen Xiao et al. (2024) | 0.693 | 0.801 | **0.315** |
| MIGE Tian et al. (2025) | 0.744 | 0.830 | 0.293 |
| Metaquery (Pan et al., 2025) | 0.737 | 0.851 | 0.301 |
| BAGEL Deng et al. (2025) | 0.777 | 0.851 | 0.307 |
| OmniGen2 Wu et al. (2025b) | 0.749 | 0.830 | 0.310 |
| **Query-Kontext** | **0.786** | **0.858** | 0.307 |

Table 7: **Quantitative results for multi-subject driven generation on Dreambench.** We highlight the **best** and second-best values for each metric.

| Method | DINO ↑ | CLIP-I ↑ | CLIP-T ↑ |
|---|---|---|---|
| *Tuning-free* | | | |
| DreamBooth Ruiz et al. (2023) | 0.430 | 0.695 | 0.308 |
| BLIP-Diffusion Li et al. (2023a) | 0.464 | 0.698 | 0.300 |
| *Specialist Models* | | | |
| Subject Diffusion Ma et al. (2024a) | 0.506 | 0.696 | 0.310 |
| MIP-Adapter Huang et al. (2024a) | 0.482 | 0.726 | 0.311 |
| MS-Diffusion Wang et al. (2025b) | 0.525 | 0.726 | 0.319 |
| UNO-FLUX Wu et al. (2025c) | **0.542** | **0.733** | 0.322 |
| *Generalist Models* | | | |
| OmniGen Xiao et al. (2024) | 0.511 | 0.722 | 0.331 |
| BAGEL Deng et al. (2025) | 0.439 | 0.683 | 0.335 |
| OmniGen2 Wu et al. (2025b) | 0.488 | 0.716 | 0.332 |
| **Query-Kontext** | 0.532 | 0.731 | **0.336** |

generation quality or photorealism. We leave this exploration in future work. For subject-driven generation on DreamBooth, Query-Kontext establishes new state-of-the-art results with DINO 0.786 and CLIP-I 0.858, significantly outperforming Metaquery (0.737 / 0.851) and UNO-FLUX (0.760 / 0.835), though with a slightly lower CLIP-T (0.307 vs. OmniGen's 0.315), as shown in Table 6. In Table 7, Query-Kontext achieves the best CLIP-T score (0.336) alongside competitive DINO (0.532) and CLIP-I results (0.731), on the multi-subject composition benchmark DreamBench.

## 4.2 QUALITATIVE RESULTS

We also provide qualitative comparisons across all task categories, including text-to-image generation, instruction editing, and customized generation, under both Chinese and English prompts. Representative examples are shown in Figure 1.

## 4.3 SHIFTED ROPE

| Method | DINO ↑ | CLIP-I ↑ | CLIP-T ↑ |
|---|---|---|---|
| w/ $src_{img}$ | 0.865 | 0.914 | 0.289 |
| w/ $ref_{img}$ | 0.786 | 0.858 | 0.307 |

We further examine the effect of the proposed shifted 2D-RoPE mechanism for handling reference images. With *source* input images, the model tends to preserve the pixel-level fidelity of the input, producing faithful reconstructions. In contrast, with *reference* input images, the model emphasizes instruction following and generalization, maintaining subject identity while generating more diverse outputs. Comparative results on the DreamBooth benchmark using source versus reference images are reported in Table 8.

Table 8: The comparison between the shifted RoPE on *source* or *reference* image.

## 4.4 QUERY–KONTEXT CONVERGENCE

In Stage 2, we analyze the convergence behavior of the diffusion model when conditioned on two settings: (i) text-only embeddings from an LLM and (ii) the mixed conditioning from both text tokens and Query–Kontext tokens generated by our fine-tuned VLM. We observe that replacing the LLM with our VLM leads to faster alignment of the diffusion model and produces superior visual results compared to the LLM-conditioned baseline, as shown in Figure 4. This demonstrates that decoupling multimodal reasoning from visual generation via Query–Kontext not only accelerates convergence but also unleashes the full potential of both the VLM and the diffusion model.

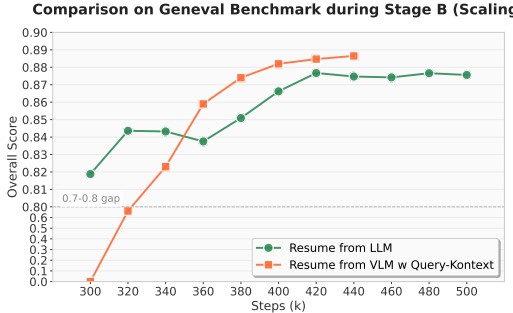

Figure 4: **Convergence validation of Query–Kontext.** Comparison on our in-house MMDiT between VLM re-alignment with Query–Kontext and LLM-based resumption.

## 5 DISCUSSION

**Economical Alignment between VLM and Diffusion Model.** Query-Kontext builds on a powerful VLM and an MMDiT-based diffusion model, leveraging the strengths of each to construct a unified multimodal-to-image generation system. The training process was conducted on 192 NVIDIA H100 GPUs (80GB), which amounts to roughly 10% of the computational resources typically required to train a large-scale diffusion model from scratch (e.g., Qwen-Image) or an integrated multimodal transformer (e.g., BAGEL). This economical alignment allows us to allocate resources more effectively, focusing on higher-level and underexplored post-training tasks such as multi-subject composition, multi-image generation, and interleaved text–image generation.

**Scaling of the Diffusion Model.** By decoupling multimodal generative reasoning in the VLM from high-fidelity visual synthesis in the diffusion model, our framework enables independent exploration on the scaling laws of each component. This separation is crucial, as VLMs and diffusion models often exhibit competing capacity requirements and benefit from different parameter budgets. In Stage 2, we attempted alignment with in-house diffusion backbones of varying sizes (0.9B, 4B, and 10B parameters). However, alignment was not always successful—particularly when employing a lightweight connector to bridge a heavy and frozen diffusion model (e.g., 10B parameters). To mitigate this, we simply unfroze the diffusion model parameters during Stage 2 training, thereby avoiding an intensive grid search over connector hyperparameters. Exploring the connector's scaling law remains an important avenue for future work.

## 6 CONCLUSION

In this work, we introduced Query-Kontext, an economical unified multimodal-to-image framework that decouples multimodal generative reasoning (handled by the VLM) from high-fidelity rendering (handled by the diffusion model). To fully harness the potential of both components, we proposed a three-stage progressive training strategy that progressively aligns the VLM with increasingly capable diffusion generators while amplifying their complementary strengths. In addition, we curated a multimodal reference-to-image dataset mixture spanning real, synthetic, and carefully filtered open-source data. Extensive experiments demonstrate that our framework achieves competitive performance across diverse tasks, including image generation, instruction editing, customized subject synthesis, and multi-subject composition.

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

APPENDIX

# A  THE USE OF LARGE LANGUAGE MODELS (LLMS)

The research ideas and experimental design of this paper were conceived entirely by the authors without the use of LLMs. During manuscript preparation, we used GPT to assist with grammar checking and refinement of language for clarity and readability.

# B  ADDITIONAL DATASET EXAMPLES

## B.1  IMAGE TRANSFORMATION

In Figure 5, we provide representative examples of naturally occurring transformation triplets constructed from large-scale web corpora. Each triplet contains a source image, an open-ended transformation instruction which is generated by Qwen2.5-VL as shown in Figure 6, and a target image. The instructions cover viewpoint changes (e.g., zoom-in/out, rotation), appearance modifications (e.g., color/material replacement), and structural adjustments (e.g., adding/removing objects). Candidate pairs are first grouped by caption proximity and then re-ranked by vision-language similarity prior to thresholding to ensure semantic alignment.

## B.2  INSTRUCTION EDITING

**Synthesis data pipeline.** To enrich the instruction editing dataset, we construct a multi-stage synthetic pipeline as illustrated in Figure 7. Given a source image, its segmentation mask, and a caption, we first extract object-level queries (e.g., "man") and generate diverse editing instructions. Specifically, large language models (Qwen2-72B) generate textual editing prompts from captions, while multimodal models (Qwen2VL-72B) leverage both captions and images to produce more fine-

A set of knitted baby accessories, including a hat and socks, with a similar light green color theme as the source image. The focus is on the accessories rather than the full body garment.

一套同样采用浅绿色主题的宝宝针织配饰，包括帽子与袜套。与源图像相同，但主要关注配饰而非完整的衣物。

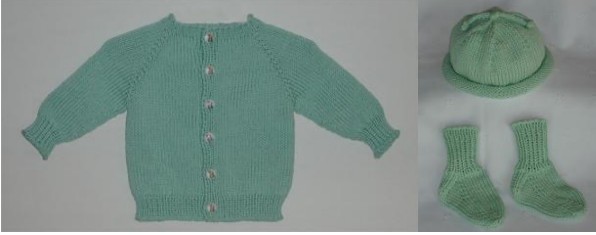

The same book cover, but from a slightly different angle, with the title and author's name more clearly visible, and the overall image slightly more tilted.

同一本书的封面，但角度略有不同，标题与作者名字更加清晰，整体图像稍微倾斜一些。

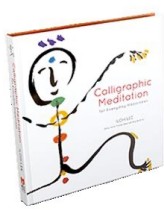 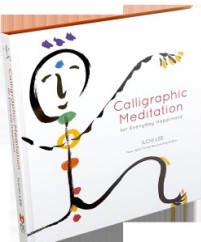

The same blue Le Creuset pot with the same design , but in a kitchen setting with additional food items like pies and soufflés around it.

相同的蓝色Le Creuset锅，具有相同的图案设计，但位于厨房环境中，并且周围有额外的食物，例如馅饼和其他甜食。

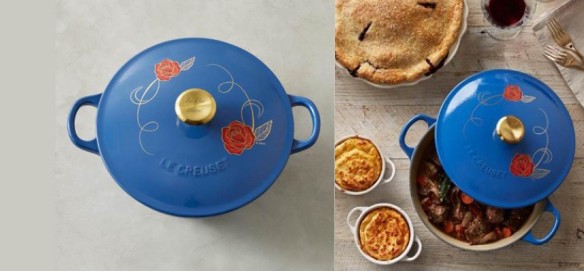

Figure 5: Examples of the image transformation task.

```
┌─ Qwen2.5-VL Prompt ──────────────────────────────────────────────────┐
│                                                                       │
│  The first image is the source image, the second image is the target  │
│  image. create an interesting text prompt that can be used with the   │
│  source images to generate the target image.                          │
│  This prompt should include:                                          │
│  - one general and unspecific similarity shared with the source image.│
│  - all differences that only the target image has.                    │
│  This prompt should NOT include:                                      │
│  - any specific details that would allow generating the target image  │
│  independently without referencing the source image.                  │
│  Remember the prompt should be concise and short (no more than 64     │
│  words).                                                              │
│                                                                       │
│  The difference should include but not limited to:                    │
│                                                                       │
│  Change in Angle. Describe the specific new angle, e.g.:              │
│  * Side view, back view, viewed from the front side                   │
│  * With a closer view, focus on the top of the item                   │
│  * Cropped in the center, in a horizontal/vertical view               │
│                                                                       │
│  Same Subject, Altered Elements. Specify added/removed elements, e.g.:│
│  * The same jacket but with a person wearing it                       │
│  * Without the package                                                │
│  * with a bowl on the right                                           │
│  * Show the engine of the same car                                    │
│                                                                       │
│  Color Change. Describe the new color(s), e.g.:                       │
│  * Blue and purple flowers instead of yellow                          │
│  * Turn the color of the vase to cyan                                 │
│  * Same design but in white                                           │
│                                                                       │
│  Position Change. Specify the new position, e.g.:                     │
│  * Put the item in the middle                                         │
│  * Move it to the side with a closer view                             │
│                                                                       │
│  Background Change. Describe the modified background, e.g.:           │
│  * With a clean background                                            │
│  * The daylight turns dim                                             │
│  * Without a background                                               │
│                                                                       │
│  Quantity Change. State the updated quantity, e.g.:                   │
│  * Show three trains of the same type                                 │
│                                                                       │
│  State/Process Change. Describe the transformation or action, e.g.:  │
│  * The beef is cooked                                                 │
│  * The man is pouring batter into a pan                               │
│  * Put it down                                                        │
│                                                                       │
│  Here are some of the example:                                        │
│  1. Prompt: The complete front view of the same jersey top.           │
│  2. Prompt: Show three blue pot with same floral design, now placed in│
│  a cozy                                                               │
│  dining scene with food, drinks, and side dishes around it.           │
│  3. Prompt: The same pair of silver rings captured in another angle   │
│  with brighter                                                        │
│  lighting and a clean white background and softer shadows.            │
│                                                                       │
│  Please generate one English prompt and one Chinese prompt, following │
│  the JSON                                                             │
│  format:                                                              │
│  ['English prompt here', 'Chinese prompt here'.]                      │
│                                                                       │
└───────────────────────────────────────────────────────────────────────┘
```

Figure 6: Example of the prompt used in Qwen2.5-VL to generate open-ended transformation instructions.

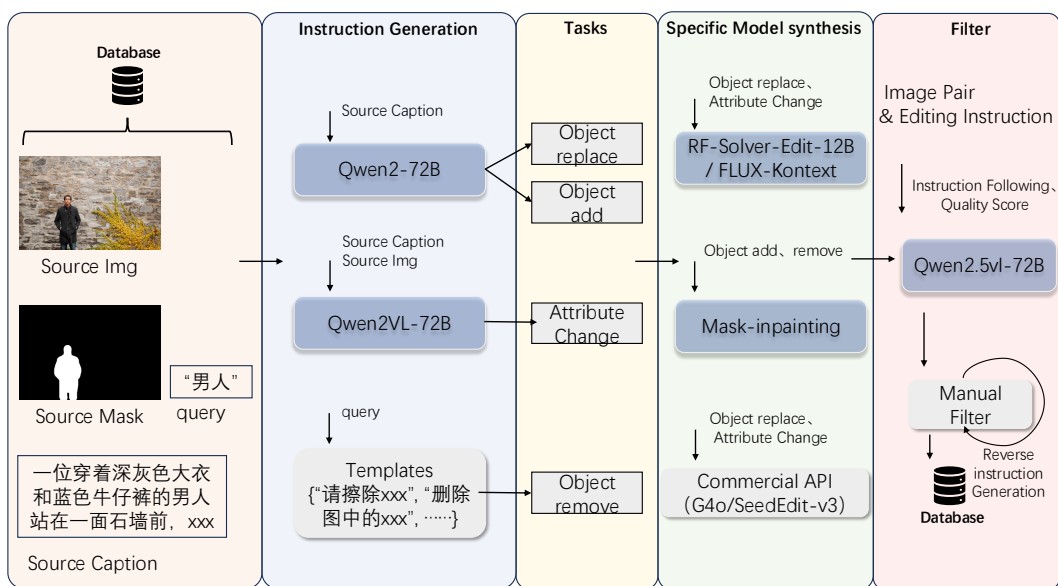

Figure 7: Examples of synthetic data pipeline for instruction Editing.

grained attribute modification instructions. In addition, we incorporate template-based instructions (e.g., "remove xxx from the image") to further handle object remove task. The generated instructions are categorized into four task types: object replacement, object addition, object removal, and attribute modification. Each task is then handled by specialized synthesis models: RF-Solver-Edit-12B Wang et al. (2024a) or FLUX-Kontext Labs et al. (2025) for replacement and attribute edits, mask-based inpainting model BlackForest (2024) for addition and removal, and commercial APIs (e.g., G4o/SeedEdit-v3) for more complex operations. Finally, the generated triplets are filtered through an automatic evaluation stage using Qwen2.5VL-72B, which scores instruction fidelity and image quality, followed by manual verification to ensure reliability. Finally, manual reverse instruction generation is applied by treating the source image as the target, ensuring supervision from authentic images without model-induced artifacts. Moreover, when applying mask-inpainting models to remove large objects, we adopt a mask augmentation strategy to mitigate the influence of shape-guided masks. Figure 8 presents a comparison between results with and without mask augmentation.

**Real video data.** Following UniReal Chen et al. (2025d), we construct image pairs from real video data to cover a broader range of non-rigid editing scenarios, and generate corresponding transformation instructions as illustrated in Figure 6. Representative data examples are provided in Figure 9.

### B.3 MULTI-SUBJECT COMPOSITION

As illustrated in Figure 10, we design a synthetic pipeline to construct high-quality multi-subject composition data. Starting from an in-house database, we combine both real and synthetic images, and generate human–object–scene lists that are further refined by large language models (LLMs) to produce natural composition instructions. Grounding-DINO Liu et al. (2023) and SAM Kirillov et al. (2023) are employed to extract object-level masks and build a mask gallery, which provides structural guidance for subsequent composition. Reference images of subjects and objects are synthesized by UNO-FLUX and GPT-Image1, while scene backgrounds are generated by mask-inpainting model. The resulting target images, together with corresponding composition instructions and scene prompts, form diverse training triplets that enhance the coverage of multi-subject scenarios.

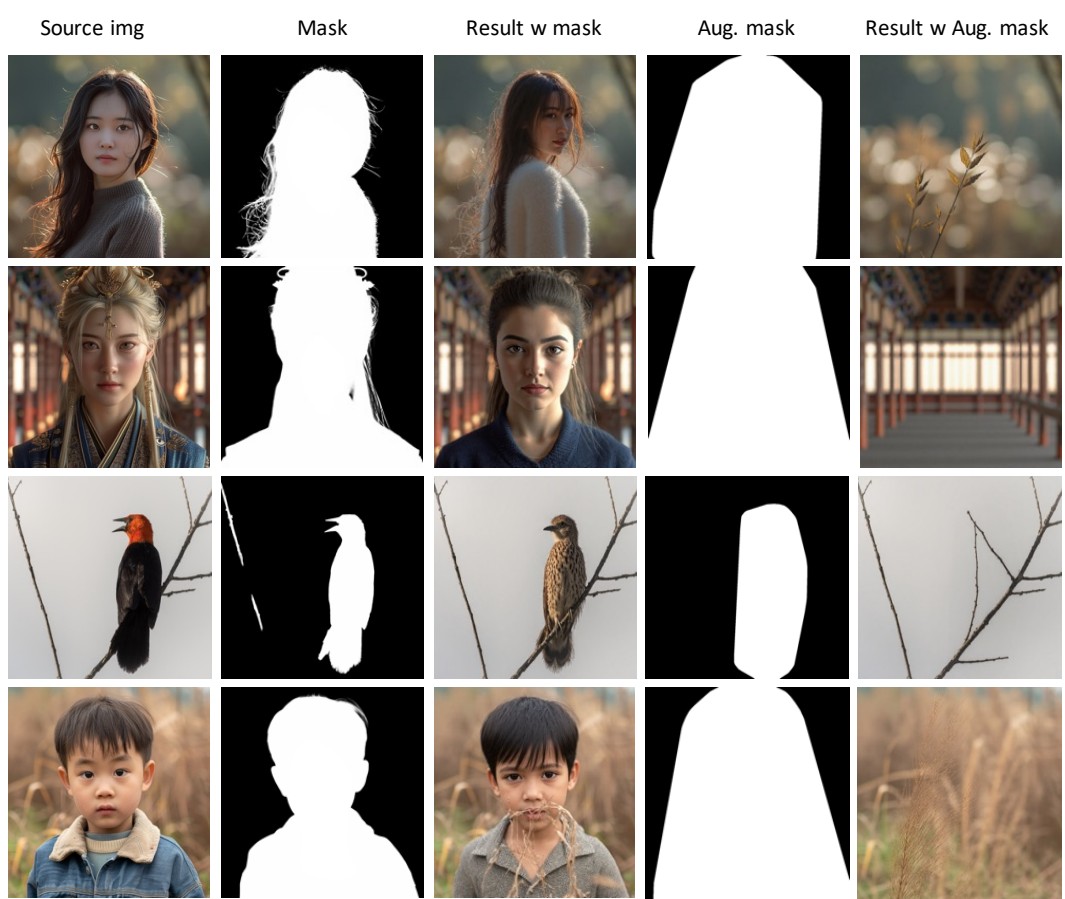

Figure 8: Examples of image inpainting with mask augmentation.

## C  ADDITIONAL ABLATION STUDIES

### C.1  QUALITATIVE COMPARISON

We include additional qualitative comparisons on dreambooth and dreambench benchmarks in Figure 11 and 12. Qualitative results demonstrate that Query-Kontext surpasses state-of-the-art UMMs (such as BAGEL Deng et al. (2025) and OmniGen2 Wu et al. (2025b)) in instruction adherence, identity fidelity, and visual harmony.

### C.2  DECOUPLING OF THE VLM AND DIFFUSION MODEL

We report the visualizations in Figure13 comparing Stage 1 and Stage 2 outputs. **Stage 1 Output:** Correct semantic layout and pose (proven reasoning) but blurry textures (limited rendering capacity). **Stage 2 Output:** The same semantic layout is maintained, but with photorealistic details (unlocked rendering capacity). This empirically validates that the VLM handles the reasoning (structure), and the Diffusion model handles the synthesis (texture), bridged effectively by the *Kontext* tokens.

### C.3  QUALITATIVE RESULTS ON UNSEEN CONCEPT GENERALIZATION AND ZERO-SHOT TASKS.

To verify the effectiveness of our reasoning-rendering decoupling in challenging scenarios, we evaluated the model on unseen concept generalization and zero-shot tasks. As illustrated in Figure 14(a) and (b), we conducted customized generation and instruction editing experiments using concepts absent from the instruction-tuning dataset, such as specific commercial brands and distinct artistic styles. Query-Kontext successfully performs complex spatial layout planning and style transfer for

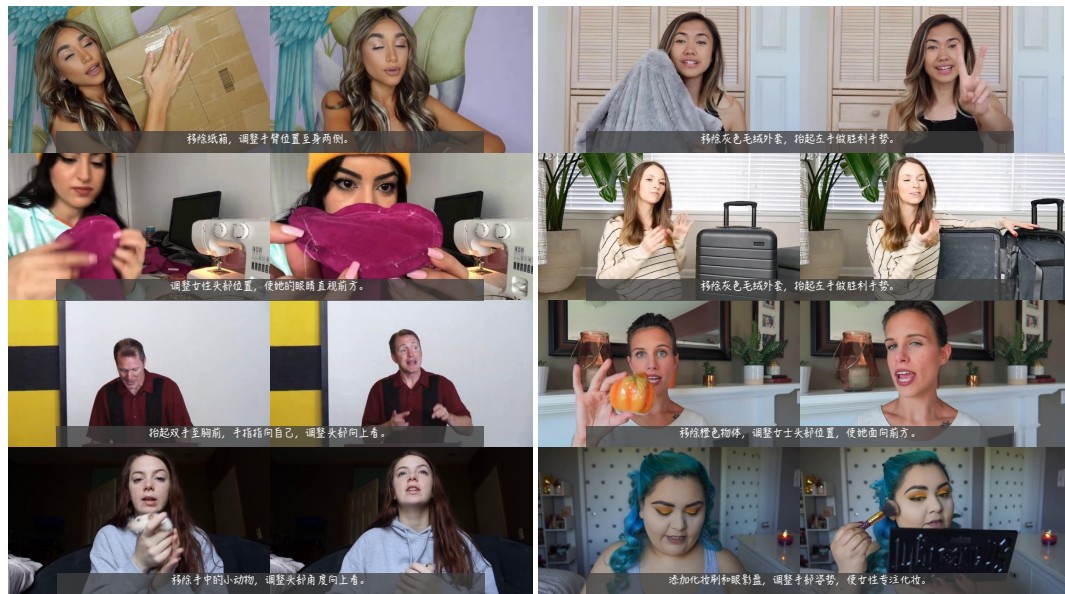

Figure 9: Examples of instruction editing data pair constructed from real video.

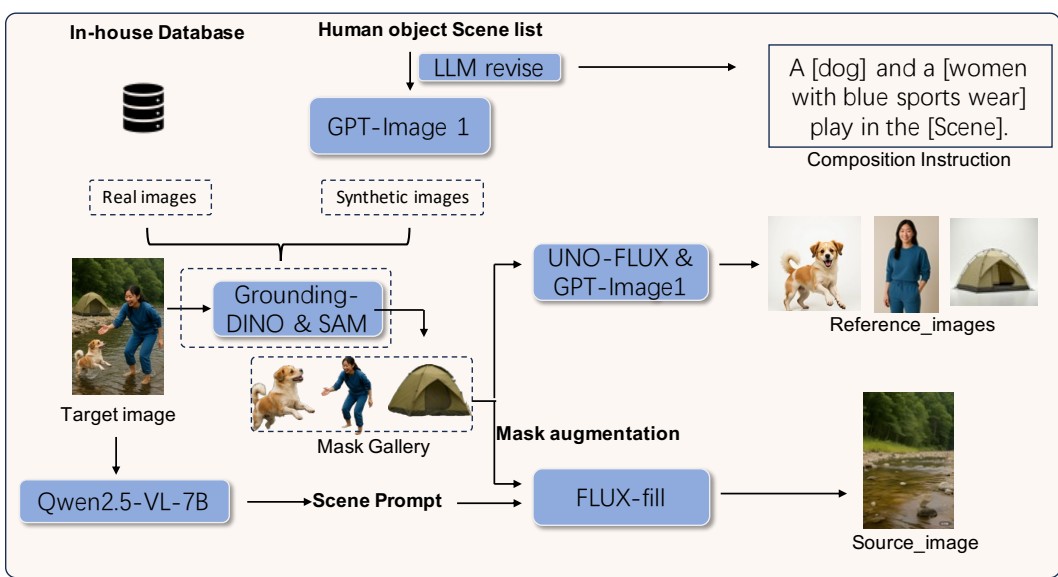

Figure 10: Examples of synthetic data pipeline for Multi-Subject Composition.

these unseen concepts. This confirms that the framework effectively leverages the "World Knowledg" inherent in the pre-trained VLM, transferring it to the diffusion head via Kontext tokens, thereby significantly outperforming baselines that rely solely on text conditioning. Furthermore, we assessed the model's generalized reasoning capabilities through a zero-shot outpainting task, as shown in Figure 14(c). Despite not being explicitly trained for outpainting, Query-Kontext correctly infers the global scene context from the reference image (e.g., harmoniously extending a portrait's body and background).

### C.4 QUALITATIVE RESULTS ON LOW-LEVEL ENCODER.

We provide comparative results of the low-level encoder in the editing task. As shown in Figure 15, incorporating the low-level encoder improves the ability to edit fine-grained image details.

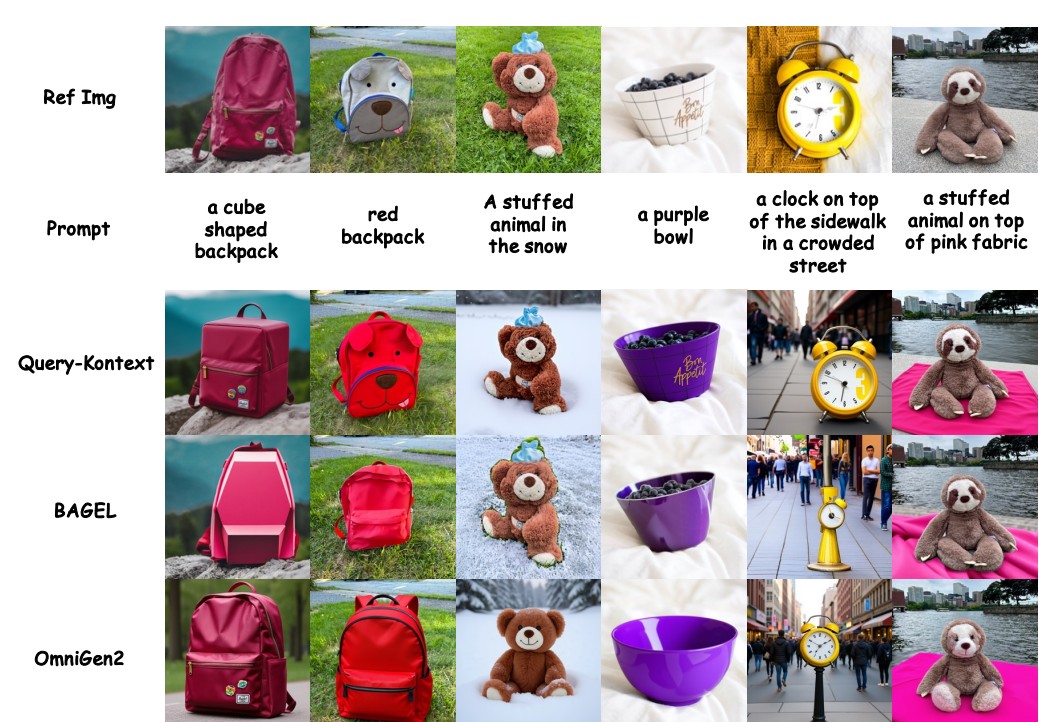

Figure 11: Qualitative Comparison on single-subject driven benchmark.

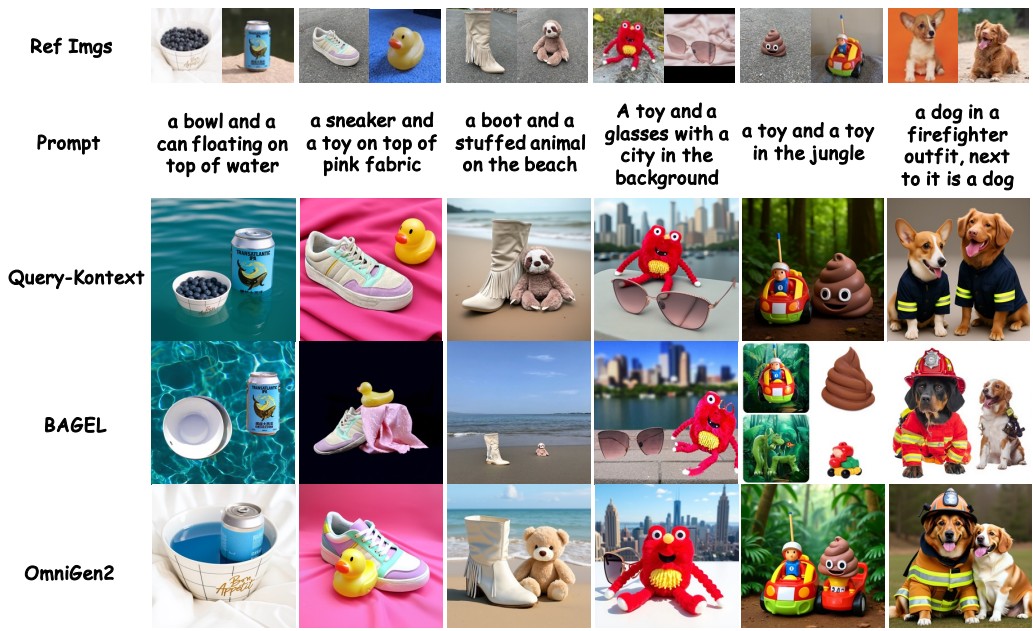

Figure 12: Qualitative Comparison on multi-subjects driven benchmark.

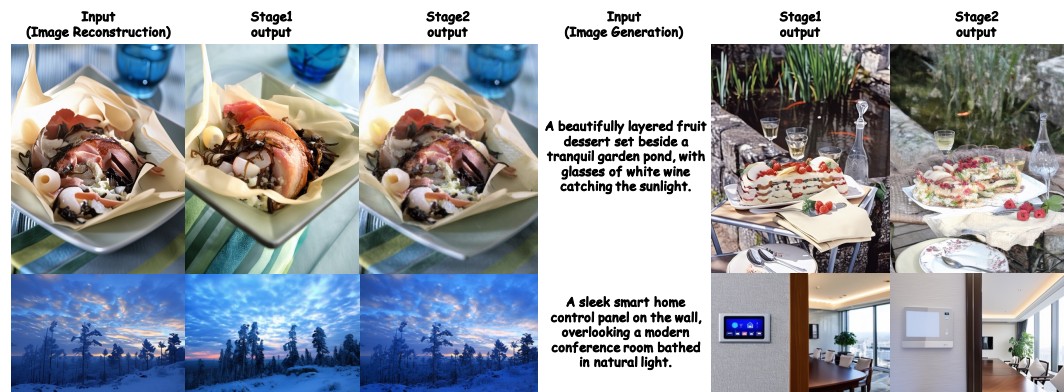

Figure 13: Qualitative comparison between the Stage 1 and Stage 2 outputs.

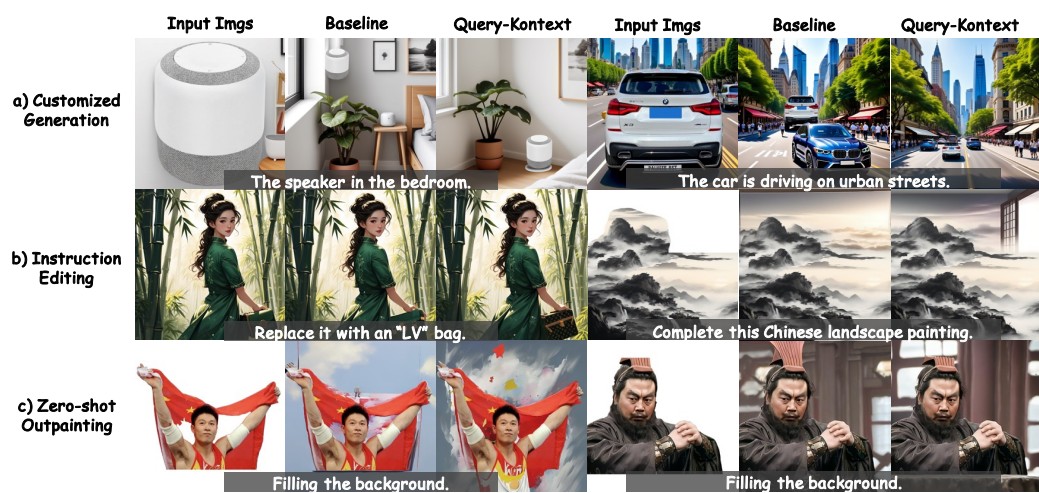

Figure 14: Qualitative results on the unseen concept generalization and zero-shot task.

## C.5 COMPARISON OF VARIOUS BRIDGE MODULES.

We empirically compared three settings (as shown in Table 9): (a) our full Query-Kontext (text + queries), (b) MLP over text tokens only, and (c) Q-Former–style queries only. With the same diffusion backbone, setting (c) is significantly harder to optimize and converges to worse GenEval performance (e.g., 0.751 vs. 0.882 at 100k steps), while our decoupled design delivers both faster convergence and better final scores.Together, these aspects make Query-Kontext more than a direct Q-Former Li et al. (2023b) adaptation. By explicitly activating the complementary capabilities of VLMs and diffusion models through this decoupled design, our approach provides a clear architectural distinction

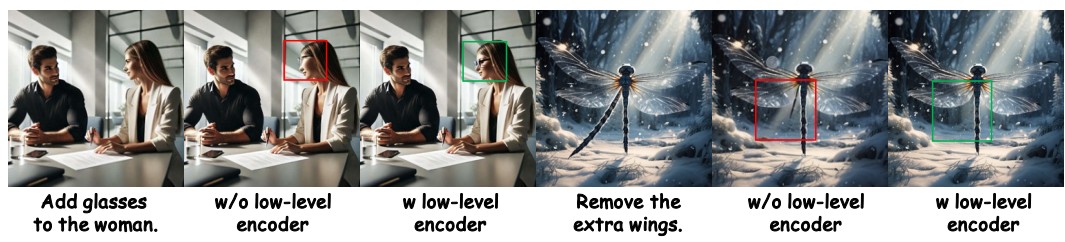

Figure 15: Qualitative comparative results for the low-level encoder.

Table 9: Comparison of different conditioning designs during **Stage 1**. Results are reported using the overall score (↑) on the **Geneval** benchmark.

| Setting | iters | 20k | 40k | 60k | 80k | 100k |
|---|---|---|---|---|---|---|
| **a. Query-Kontext** | Text (512) & query (128) | **0.661** | **0.823** | **0.859** | **0.874** | **0.882** |
| **b. MLP** | only Text (512) | 0.587 | 0.67176 | 0.757 | 0.770 | 0.814 |
| **c. Qformer** | only query (640) | 0.359 | 0.596 | 0.689 | 0.732 | 0.751 |

Table 10: Ablations on positional encoding, number of reference images, and LoRA ranks.

| Setting | DINO | CLIP-I | CLIP-T |
|---|---|---|---|
| LoRA $r=64$ | 0.752 | 0.841 | 0.298 |
| LoRA $r=128$ | 0.786 | 0.858 | 0.307 |
| LoRA $r=256$ | 0.777 | 0.834 | 0.311 |

Table 11: Configuration of Query-Kontext architecture.

| Configuration | MLLM | | VAE | | Connector | MMDiT |
|---|---|---|---|---|---|---|
| | ViT | LLM | Enc | Dec | | |
| # Layers | 32 | 28 | 8 | 14 | 2 | 42 |
| # Num Heads (Q / KV) | 16 / 16 | 28 / 4 | - | - | - | 40 / 40 |
| Head Size | 80 | 128 | - | - | - | 64 |
| Intermediate Size | 3,456 | 18,944 | - | - | - | 10240 |
| Patch / Scale Factor | 14 | - | 8x8 | 8x8 | - | 2 |
| Channel Size | - | - | 16 | 16 | - | - |
| # Parameters | 7B | | 34M | 50M | 5.9M | 10B |

specifically tailored for controllable and stable diffusion conditioning, distinguishing it from both Q-Former–style bridges and existing UMMs.

### C.6 LoRA Rank

We evaluate LoRA ranks $\{64, 128, 256\}$ on both the diffusion model and the MLLM adapters, observing faster convergence at higher ranks with marginal quality gains beyond $r=128$.

## D IMPLEMENTATION

**Connector.** A two-layer MLP maps text and kontext tokens into the diffusion latent space; the outputs are concatenated before being fed to the diffusion transformer.

**Shifted 2D-RoPE (sRoPE).** Let the latent spatial size be $h \times w$. For the $n$-th reference image latent, we shift coordinates to the positive quadrant:

$$(i_{\text{ref}}^n, j_{\text{ref}}^n) = (i + w \cdot n, \ j + h \cdot n), \quad i \in [0, w-1], \ j \in [0, h-1], \ n \in [1, N], \tag{3}$$

while for the source image latent we shift to the negative quadrant:

$$(i_{\text{src}}', j_{\text{src}}') = (-i, \ -j), \quad i \in [0, w-1], \ j \in [0, h-1]. \tag{4}$$

The shifted rotary embeddings are then applied at the corresponding coordinates to both input-image latents and the noisy latent of the current diffusion step, enabling disambiguation between identity-preserving references and pixel-faithful sources.

**Infrastructure.** We adopt a hybrid parallel optimization strategy during training. we enable tensor parallelism on the VLM side. For the diffusion model, we use parameter sharding (ZeRO Stage-2) together with bfloat16 (BF16) mixed-precision training. To keep sequence lengths uniform within a

mini-batch, we maintain two independent bucketeers—by image aspect ratio (supporting 1:1, 1:2, 2:3, 3:4, 3:5, 4:5, and 9:16) and by the number of reference images—so that samples in the same batch produce the same number of latent tokens, reducing padding and improving throughput.

# E RELATED WORK

## E.1 INSTRUCTION-BASED IMAGE EDITING

Early diffusion-based editing splits into *training-free* and *training-based* approaches. Training-free methods manipulate the denoising trajectory via inversion Meng et al. (2021); Mokady et al. (2023); Kawar et al. (2023); Couairon et al. (2022); Wallace et al. (2023); Chen et al. (2023b) or attention control Cao et al. (2023); Hertz et al. (2022); Parmar et al. (2023) and require no additional training, but often struggle with fine-grained instruction fidelity and identity preservation. InstructPix2Pix Brooks et al. (2023) pioneered training-based approaches Chen et al. (2023b); Zhao et al. (2024b); Li et al. (2024b); Yu et al. (2025); Zhang et al. (2025) by finetuning a pretrained diffusion backbone on curated (image, instruction, edited-image) triplets, yielding stronger instruction following and higher fidelity. More recently, a trend towards tighter integration of *multimodal* understanding and generation has emerged to empower more complex editing instructions. Works like SmartEdit Huang et al. (2024b) and Step1X-Edit Liu et al. (2025b) leverage MLLM latent representations to guide structured or latent-conditioned editing, ACE Han et al. (2024), ACE++ Mao et al. (2025) and FLUX.1 Kontext Labs et al. (2025) integrate text and image context for instruction-guided editing. UniVG Fu et al. (2025), SeedEdit 3.0 Wang et al. (2025a), and Qwen-Image Wu et al. (2025a) demonstrate generalist architectures capable of diverse tasks while preserving identity and fidelity.

## E.2 UNIFIED MULTIMODAL MODELS

Unified Multimodal Models (UMMs) have recently attracted significant attention for their ability to unify both understanding and generation within a single architecture. Existing approaches can be broadly categorized into two strategies. The first strategy develops *native* UMMs Team (2024); Zhou et al. (2024); Xie et al. (2024; 2025); Wu et al. (2024); Chen et al. (2025e); Ma et al. (2024b); Wang et al. (2024e); Tong et al. (2024); Deng et al. (2025), which are trained to fuse multimodal understanding and generation capabilities at the early stage, usually involving autoregressive or diffusion modeling. While conceptually elegant, they often present considerable challenges in training and scaling. The second strategy *assembles* unified frameworks Pan et al. (2025); Wu et al. (2025a); Chen et al. (2025a); Labs et al. (2025); Liu et al. (2025a); Chen et al. (2025d); Fu et al. (2025); Chen et al. (2025c) by coupling existing vision-language models (VLMs)Bai et al. (2025); Wang et al. (2024c) for understanding with powerful diffusion-based generatorsEsser et al. (2024a); William & Xie (2023); Labs (2024). This is typically achieved through learnable tokens or tuning adapters. Our work builds on this line of research, introducing a more refined mechanism for cross-modal representation fusion and controllable generation.

## E.3 EDITING DATA CURATION

High-quality and diverse datasets of ⟨original image, instruction, edited image⟩ triplets are fundamental for training powerful editing models. MagicBrush Zhang et al. (2023) represents the manual annotation approach. InstructPix2Pix Brooks et al. (2023) pioneered data synthesis by using GPT-3 Brown et al. (2020) and Prompt-to-Prompt Hertz et al. (2022). To improve quality, HIVE Zhang et al. (2024) introduced human feedback for quality assessment and training. HQ-Edit Hui et al. (2024) and UltraEdit Zhao et al. (2024b) scaled up dataset size and difficulty using more powerful models like GPT-4V OpenAI. (2023) and DALL-E 3 Betker et al. (2023), along with fine-grained annotations. SEED-Data-Edit Ge et al. (2024) enhances diversity through re-generation and re-annotation techniques, while SeedEdit 3.0 Wang et al. (2025a) systematically upgrades both data sources and data merging. More recently, NHR-Edit Kuprashevich et al. (2025b) automates the mining of high-quality triplets from powerful open-sourced generative models like FLUX Labs (2024), reducing manual effort and improving data realism.

