# OpenReview forum: "Query-Kontext: An Unified Multimodal Model for Image Generation and Editing"
_ICLR.cc/2026/Conference — Submitted to ICLR 2026_

### Official Review · Reviewer_Asae · 2025-10-28

**Soundness:** 3
**Presentation:** 2
**Contribution:** 3
**Rating:** 6
**Confidence:** 3

**Summary:**

Query-Kontext is a unified multimodal model (UMM) for text-to-image generation (T2I) and editing (TI2I) that decouples multimodal generative reasoning in vision-language models (VLMs) from high-fidelity synthesis in diffusion models. It bridges them via "kontext" tokens encoding semantic cues and coarse image conditions from multimodal inputs. A three-stage progressive training strategy is introduced: (1) connecting VLM to a lightweight diffusion head using LoRA; (2) scaling to a large pre-trained diffusion model for enhanced realism; (3) adding a low-level image encoder for fidelity and instruction tuning. A comprehensive dataset integrates real, synthetic, and open-source data across diverse tasks. Experiments demonstrate it matches strong UMM baselines and outperforms task-specific SOTA methods in several cases.

**Strengths:**

* The paper introduces Query-Kontext, which innovatively bridges VLMs and diffusion models using "kontext" tokens that encode semantic cues and coarse image conditions. This decouples complex generative reasoning from high-fidelity synthesis, addressing limitations in existing unified frameworks where these capabilities are entangled.
* The proposed shifted 2D Rotary Position Embedding distinguishes source and reference images by offsetting coordinates, preventing confusion in multi-image tasks like composition and editing.
* Experiments demonstrate Query-Kontext matches strong unified baselines and outperforms task-specific SOTAs in several cases across T2I, TI2I, customization, and composition.

**Weaknesses:**

* The definition and role of kontext tokens are a little bit vague and hard to follow, like the fixed-length kontext tokens (Q) are central to bridging VLM and diffusion, yet the length K and how to encode "coarse image conditions" are not specified or ablated.
* The three-stage training and shifted RoPE are central to decoupling reasoning and synthesis, but no ablations validate their contributions—e.g., no experiments showing performance drop without stage 3's low-level encoder or without shifted RoPE in multi-subject tasks. This obscures why the design achieves stated goals.
* The data curation heavily skews toward in-house Chinese image-text pairs, compared to only 30M open-source English pairs. This increases my concern about the fairness in comparative evaluation and the soundness of empirical validity.

**Questions:**

1. Stage 3 introduces a low-level encoder for fine-grained cues. Is this a standard SD-VAE or a custom implementation? How does it interact with kontext tokens—does it override or complement them?
2. The shifted Rotary Position Embedding (sRoPE) is an interesting addition for handling multiple reference images and source images by shifting coordinates into positive/negative quadrants. How does this interact with varying image resolutions or aspect ratios?

---

> ### Author Response · Authors · 2025-11-27
> **Response to Reviewer Asae (1/2)**
>
> Dear Reviewer Asae:
>
> **W1: The definition and role of kontext tokens are a little bit vague and hard to follow, like the fixed-length kontext tokens (Q) are central to bridging VLM and diffusion, yet the length K and how to encode "coarse image conditions" are not specified or ablated.**
>
> We apologize for the lack of clarity and we explicitly define that the *Kontext* tokens (Q) consist of a fixed sequence of $K=128$ learnable tokens, denoted as $Q=\{q_1, ..., q_K\}$.
> Within the VLM, these tokens are concatenated with text and reference image tokens and processed via self-attention, allowing $Q$ to aggregate and fuse semantic and visual reference information.
>
> Functioning as a coarse signal carrier during Stage 1, the VLM is trained to compress reasoning and structural planning into $Q$, enabling a *lightweight* diffusion head to decode them into a target image. To validate this role, we present a qualitative comparison in **Figure 13** (Appendix). We observe that the Stage 1 output (conditioned solely on $Q$) exhibits correct semantic structure and pose but lacks high-frequency detail, confirming that $Q$ successfully encodes the essential "coarse visual signal" for reasoning and planning. Subsequently, the Stage 2 output demonstrates how the full-scale MMDiT effectively refines this coarse signal into a high-fidelity generation.
>
> **W2: The three-stage training and shifted RoPE are central to decoupling reasoning and synthesis, but no ablations validate their contributions—e.g., no experiments showing performance drop without stage 3's low-level encoder or without shifted RoPE in multi-subject tasks. This obscures why the design achieves stated goals.**
>
> We agree that the necessity of sRoPE warrants stronger evidence. Standard RoPE assigns identical positional indices to reference latents and noisy latents, causing severe spatial aliasing. To demonstrate this, we introduce new ablation results on multi-subject generation alongside the single-subject results based on Table 8.
>
> **Table 6: The comparison between the shifted RoPE and regular RoPE on single-subject image generation.**
>
> | Method | DINO ↑ | CLIP-I ↑ | CLIP-T ↑ |
> | :--- | :--- | :--- | :--- |
> | Regular RoPE | 0.757 | 0.835 | 0.301 |
> | shifted RoPE (w/ $ref_{img}$) | 0.786 | 0.858 | 0.307 |
>
> In addition, we provide new **Multi-Subject** ablation results below. The degradation with standard RoPE is pronounced in multi-subject scenarios, confirming that sRoPE is essential for disambiguating reference tokens from generation targets.
>
> **Table 7: The comparison between the shifted RoPE and regular RoPE on multi-subject image generation.**
>
> | Method | DINO ↑ | CLIP-I ↑ | CLIP-T ↑ |
> | :--- | :--- | :--- | :--- |
> | Regular RoPE | 0.426 | 0.589 | 0.311 |
> | shifted RoPE (w/ $ref_{img}$) | 0.532 | 0.731 | 0.336 |
>
> On the other hand, the low-level encoder follows established designs (e.g., Step1X-Edit, Uniworld) to inject fine-grained cues that the coarse $Q$ tokens might miss; thus, it is not part of our core contribution. Nevertheless, we include qualitative comparisons in the revision (**Figure 15**) to show that the low-level encoder improves fine-detail referencing and increases the editing success rate.

---

> > ### Author Response · Authors · 2025-11-27
> > **Response to Reviewer Asae (2/2)**
> >
> > **W3: The data curation heavily skews toward in-house Chinese image-text pairs, compared to only 30M open-source English pairs. This increases my concern about the fairness in comparative evaluation and the soundness of empirical validity.**
> >
> > We address the fairness and validity concerns through a rigorous evaluation protocol and by leveraging the inherent properties of our data. As noted in lines 421-423, for benchmarks like GenEval, we utilized DeepSeek to rewrite prompts into Chinese, ensuring that the comparison accurately reflects the model's generative capability rather than language misalignment. Furthermore, since modern VLMs possess strong inherent cross-lingual transfer abilities, our experiments demonstrate that the model generalizes well even with a smaller proportion of English data. Finally, our "Chinese" dataset is linguistically diverse in content; it contains a vast amount of English text-rendering tasks **(e.g., prompts like "一块黑板上面印有'Happy New Year'")** which are categorized under Chinese data sources but effectively train English text rendering capabilities.
> >
> > **Q1: Stage 3 introduces a low-level encoder for fine-grained cues. Is this a standard SD-VAE or a custom implementation? How does it interact with kontext tokens—does it override or complement them?**
> >
> > Regarding the Stage 3 low-level encoder, we implement it as a standard SD-VAE (with 8x downsampling and 16 channels) that is further fine-tuned. In terms of interaction, it **complements** the *Kontext* tokens rather than overriding them. Specifically, the low-level features are concatenated with the noisy latents and interact with $Q$ via self-attention within the MMDiT. In this synergy, $Q$ provides the high-level semantic plan, while the encoder provides the necessary structural and textural guidance.
> >
> > **Q2: The shifted Rotary Position Embedding (sRoPE) is an interesting addition for handling multiple reference images and source images by shifting coordinates into positive/negative quadrants. How does this interact with varying image resolutions or aspect ratios?**
> >
> > We handle varying image resolutions and aspect ratios via a resolution bucketing strategy detailed in Appendix (lines 1327-1332). During training, we support 7 distinct aspect ratios (1:1, 1:2, 2:3, 3:4, 3:5, 4:5, and 9:16). Within each batch, while the source, reference, and target images share the same token count, they can form $7 \times 7 \times 7 = 343$ different resolution combinations. Because the model is trained on this dense combinatorial grid using sRoPE, it generalizes robustly to varying resolutions and aspect ratios during inference without suffering from coordinate misalignment.
> >
> > Best regards,
> >
> > Authors of Paper 2376

---

> > > ### Comment · Reviewer_Asae · 2025-11-28
> > >
> > > As for W3, I am more concerned—compared to the language issue—that the large proportion of *in-house* data may compromise the fairness in comparative evaluation.

---

> > > > ### Author Response · Authors · 2025-11-29
> > > > **Official Comment by Authors**
> > > >
> > > > Thanks for raising this concern. To achieve competitive results against commercial black-box models and state-of-the-art proprietary systems, utilizing a large-scale in-house dataset was necessary and common in practice. Moreover, We are currently undergoing the internal approval process to open-source the curated dataset and the synthetic data generation pipeline, ensuring future reproducibility.
> > > >
> > > > We hope these clarifications have resolved your concerns. Thank you once again for your thoughtful review of our submission.
> > > >
> > > > Best regards,
> > > >
> > > > The Authors

---

> > > ### Comment · Reviewer_Asae · 2025-11-28
> > >
> > > Thank you, the clarifications on Q1 and Q2 are helpful, and I no longer have any concerns about them.

---

### Official Review · Reviewer_fgrd · 2025-10-29

**Soundness:** 3
**Presentation:** 3
**Contribution:** 4
**Rating:** 6
**Confidence:** 4

**Summary:**

This paper proposes Query-Kontext, a unified multimodal framework designed to decouple multimodal generative reasoning (handled by a Vision-Language Model, VLM) from high-fidelity visual synthesis (handled by a diffusion model).
The key innovation is the introduction of a multimodal “kontext” — a set of semantic and coarse-grained image tokens bridging the VLM and diffusion backbone.
A three-stage progressive training strategy is developed:
	1.	Stage 1: Fine-tune a VLM with LoRA adapters to generate kontext tokens and align with a lightweight diffusion head.
	2.	Stage 2: Scale to a large pre-trained diffusion transformer (MMDiT) and align kontext/text embeddings through text-to-image and reconstruction tasks.
	3.	Stage 3: Introduce a low-level image encoder for fine-grained fidelity and identity preservation, while keeping the VLM frozen.
A comprehensive multimodal dataset (≈200 M samples) mixing real, synthetic, and curated open-source sources is built, spanning text-to-image generation, instruction-based editing, customized subject generation, and multi-subject composition.
Experiments show that Query-Kontext achieves state-of-the-art or competitive performance on GenEval, GEdit-Bench, DreamBooth, and DreamBench, outperforming several unified or task-specific baselines such as BAGEL, Qwen-Image, and GPT-Image in multiple metrics.

**Strengths:**

1. Elegant decoupling of reasoning (VLM) and rendering (diffusion) via “kontext” tokens.
2. Extensive dataset curation unifying text-to-image, editing, and composition.
3. Strong empirical performance across multiple public benchmarks (GenEval 0.88; GEdit-Bench 7.66/7.65; DreamBooth DINO 0.786).
4. Comprehensive analysis including ablation, convergence, etc.

**Weaknesses:**

1. Although Query-Kontext presents a solid and industrial-grade unified multimodal system,  the architectural novelty of Query-Kontext is limited — aside from the sRoPE design, the framework closely mirrors existing unified models such as Qwen-Image in structure and modality bridging.
2. The experiments cover standard benchmarks but lack targeted evaluation on compositional reasoning or unseen-concept editing, which are the key scenarios where the claimed reasoning–rendering decoupling should show advantages.
3. The paper failed to present the improvements of the proposed framework compared with the inhouse MMDiT.
4. The caption of table 2 should be on top of the table.

**Questions:**

See weeknesses

---

> ### Author Response · Authors · 2025-11-27
> **Response to Reviewer fgrd**
>
> Dear Reviewer fgrd:
>
> We thank the reviewer for the encouraging feedback, highlighting our elegant decoupling design, extensive dataset curation, and strong empirical performance across benchmarks. We address the concerns below.
>
> **W1: Although Query-Kontext presents a solid and industrial-grade unified multimodal system, the architectural novelty of Query-Kontext is limited — aside from the sRoPE design, the framework closely mirrors existing unified models such as Qwen-Image in structure and modality bridging.**
>
> While we acknowledge that recent concurrent UMMs, such as Qwen-Image [1], Uniworld [2], and Metaquery [3], converge on similar high-level components (VLM encoders + VAE), our approach offers distinct architectural and methodological contributions. Architecturally, we introduce Kontext tokens as an explicit bridge that disentangles multimodal reasoning from high-fidelity rendering, rather than merging heterogeneous signals into a single latent bottleneck. This design strictly separates textual semantics, query-based contextual cues, and low-level visual features, enabling the VLM and diffusion model to specialize in complementary roles without interference. Beyond architecture, our core novelty lies in a three-stage progressive training curriculum that economically "glues" pre-trained components. This decoupled strategy effectively mitigates the capacity competition between reasoning and rendering. Combined with our specific Shifted-RoPE design, this framework allows Query-Kontext to excel across a broader spectrum of different tasks, such as instruction editing and subject customization, which purely unified baselines often struggle to balance simultaneously.
>
> **W2: The experiments cover standard benchmarks but lack targeted evaluation on compositional reasoning or unseen-concept editing, which are the key scenarios where the claimed reasoning–rendering decoupling should show advantages.**
>
> **W3: The paper failed to present the improvements of the proposed framework compared with the inhouse MMDiT.**
>
> We thank the reviewer for these insightful comments. We agree that these scenarios are critical for verifying our reasoning-rendering decoupling and demonstrating improvements over the backbone. To address this, we have added new experiments and visualizations in the Appendix (Sec C), comparing Query-Kontext against a strong baseline where our in-house MMDiT is directly Supervised Fine-Tuned (SFT) on the same downstream tasks. First, regarding **Unseen-Concept Editing & Customization**, we evaluate the model on concepts absent from our instruction-tuning dataset, such as specific commercial brands and artistic styles, as shown in the new **Figure 14(a)** and **Figure 14(b)**. While the baseline struggles to ground these unknown entities, Query-Kontext successfully performs complex spatial layout planning and style transfer. This confirms that our framework effectively leverages the "World Knowledge" inherent in the pre-trained VLM by transferring it to the diffusion head via Kontext tokens, thereby significantly outperforming baselines that rely solely on text conditioning. Second, we tested generalized reasoning via a **Zero-Shot Outpainting** task in **Figure 14(c)**. Despite not being explicitly trained for outpainting, Query-Kontext correctly infers the scene context from the reference image, for instance by harmoniously extending a portrait's body and background.
>
> **W4: The caption of table 2 should be on top of the table.**
>
> Thank you for pointing this out. We have corrected the formatting in the revised manuscript.
>
> Best regards,
>
> Authors of Paper 2376
>
> [1] Wu, C., et al. (2025). "Qwen-Image Technical Report." arXiv preprint.
>
> [2] Lin, B., et al. (2025). "UniWorld-V1: High-Resolution Semantic Encoders for Unified Visual Understanding and Generation." arXiv preprint.
>
> [3] Pan, X., et al. (2025). "Transfer between Modalities with MetaQueries." arXiv preprint.

---

> ### Comment · Reviewer_fgrd · 2025-11-28
> **Thank you for your rebuttal.**
>
> Dear Authors,
>
> Thank you for your rebuttal. I believe this is a large-scale and meaningful contribution, and I hope the authors can release their code, model and data soon for the community. I will keep my positive score.
>
> Regards,
>
> Reviewer fgrd

---

> > ### Author Response · Authors · 2025-11-28
> > **Official Comment by Authors**
> >
> > We sincerely appreciate your positive feedback and your recognition of the significance and scale of our contribution.
> > We are actively working on the release process for the code, models, and data. Please feel free to let us know if you have any further questions regarding data curation, infrastructure details, or reproducibility during the discussion period.
> >
> > Best regards,
> >
> > The Authors

---

### Official Review · Reviewer_Uvkf · 2025-10-31

**Soundness:** 2
**Presentation:** 2
**Contribution:** 2
**Rating:** 4
**Confidence:** 4

**Summary:**

Query-Kontext try to answer an critical question: can we decouple multimodal generative reasoning from high-fidelity visual synthesis, while still maintaining (or even improving) overall performance? To achieve this goal, the authors design an architecture that physically and functionally separates the reasoning and rendering processes. And, three-stage progressive training gradually enforces this separation.  After well trained on large-scale multi-modal datasets, it achieves competitive results compared to existing methods.

**Strengths:**

1. By applying clear modular decoupling, explicitly separates reasoning (semantic understanding, grounding) from rendering (fidelity and realism);
2. Achieves comparable results on both unified and task-specific models across T2I, editing, and personalization;

**Weaknesses:**

1. Query-Kontext shows that reasoning and synthesis can be effectively decoupled. But they don’t fully explain why this decoupling works, so the answer is practically convincing but theoretically shallow.
2. It is unclear the effects of shifted RoPE compared to the standard RoPE.
3. The core in-house dataset (e.g., 170M+ Chinese pairs, synthetic triplets) is not public, limiting reproducibility.
4. The connector module struggles with aligning large (10B) frozen diffusion models, forcing full fine-tuning — potentially undermining modularity claims.
5. Mostly quantitative; limited qualitative ablation or human preference studies.
6. There are format issues in the cited references.

**Questions:**

See the weaknesses.

**Details Of Ethics Concerns:**

There are in-house datasets (e.g., 170M+ Chinese pairs, synthetic triplets) is not public, limiting reproducibility.

---

> ### Author Response · Authors · 2025-11-27
> **Response to Reviewer Uvkf (1/2)**
>
> Dear Reviewer Uvkf:
>
> We thank the reviewer for the thoughtful and detailed feedback. Below we address each concern point-by-point.
>
> **W1: Query-Kontext shows that reasoning and synthesis can be effectively decoupled. But they don’t fully explain why this decoupling works, so the answer is practically convincing but theoretically shallow.**
>
> We appreciate the reviewer **prompting** us to provide a deeper theoretical justification. The effectiveness of our decoupling strategy is not accidental but grounded in two complementary deep learning principles: **Frequency-based Feature Decomposition** (explaining *why* we decouple) and the **Information Bottleneck Principle** (explaining *how* we enforce it).
>
> **1. Why Decouple: Alignment of Inductive Biases and Frequency Domains.**
> Theoretically, image generation involves two distinct modalities of information:
> *   **Low-Frequency Semantics (Reasoning):** Global structure, spatial layout, and object consistency. Transformers (LLMs/VLMs) possess a strong inductive bias for capturing long-range dependencies, mathematically acting as low-pass filters that prioritize global shape and semantic logic over high-frequency noise [1].
> *   **High-Frequency Details (Synthesis):** Local textures, lighting, and pixel-level realism. Diffusion models, operating as iterative denoisers, excel at refining these continuous, high-frequency signals [2].
>
> In Naive UMMs, forcing a single model to handle both creates a conflict in inductive biases. Our architecture explicitly decouples these tasks: the VLM generates the **"Semantic Blueprint"** (low-frequency), while the Diffusion model acts as the **"Detail Renderer"** (high-frequency). This separation ensures that "reasoning" does not compete with "Synthesis" for model capacity.
>
> **2. How Decouple: Enforcing Reasoning via Information Bottlenecks.**
> This separation is achieved under the **Information Bottleneck (IB)** principle [3, 4].
>
> In **Stage 1**, we deliberately connect the VLM to a *lightweight* diffusion head with severely limited capacity. Since this lightweight head lacks the parameter budget to store complex knowledge or infer spatial logic independently, it acts as a functional constraint. To minimize the diffusion loss, the gradient flow compels the VLM to compress all necessary structural and reasoning information into the *Kontext* tokens ($Q$). Consequently, the tokens $Q$ become a maximally informative representation of the target image's structure, leaving the head to perform only basic reconstruction. This design effectively prevents the "shortcut learning" often seen in large end-to-end models, ensuring that the reasoning capability is isolated and fully developed within the VLM before the heavy renderer (MMDiT) is introduced in Stage 2.
>
> **Qualitative Evidence of Theory:**
> We have added visualizations in **Figure 13** (Appendix) comparing Stage 1 vs. Stage 2 outputs. **Stage 1 Output:** Correct semantic layout and pose (proven reasoning) but blurry textures (limited rendering capacity). **Stage 2 Output:** The same semantic layout is maintained, but with photorealistic details (unlocked rendering capacity).
> This empirically validates our theoretical claim: the VLM handles reasoning (structure), and the Diffusion model handles synthesis (texture), bridged effectively by the *Kontext* tokens.
>
> **W2: It is unclear the effects of shifted RoPE compared to the standard RoPE.**
>
> We thank the reviewer for noting this issue and now clarify the effect of shifted RoPE.
> Standard RoPE assigns identical positional indices to the reference-image latents and the noisy latents, which leads to severe spatial confusion.
> Based on Table 8, we additionally report single- and multi-subject results as follows:
>
> **Table 4: The comparison between the shifted RoPE and regular RoPE on single-subject image generation.**
>
> | Method | DINO ↑ | CLIP-I ↑ | CLIP-T ↑ |
> | :--- | :--- | :--- | :--- |
> | Regular RoPE | 0.757 | 0.835 | 0.301 |
> | shifted RoPE (w/ $ref_{img}$) | 0.786 | 0.858 | 0.307 |
>
> **Table 5: The comparison between the shifted RoPE and regular RoPE on multi-subject image generation.**
>
> | Method | DINO ↑ | CLIP-I ↑ | CLIP-T ↑ |
> | :--- | :--- | :--- | :--- |
> | Regular RoPE | 0.426 | 0.589 | 0.311 |
> | shifted RoPE (w/ $ref_{img}$) | 0.532 | 0.731 | 0.336 |
>
> The degradation with standard RoPE is even more pronounced in multi-subject scenarios. These results confirm that shifted RoPE is essential for disambiguating reference-image tokens from noisy latents.
>
> [1] Ghiasi, A., et al. (2022). "What do vision transformers learn? A visual exploration." arXiv preprint.
>
> [2] Si, C., et al. (2024). "FreeU: Free Lunch in Diffusion U-Net." CVPR.
>
> [3] Alemi, A. A., Fischer, I., Dillon, J. V., & Murphy, K. (2017). Deep Variational Information Bottleneck. ICLR.
>
> [4] Tishby, N., Pereira, F. C., & Bialek, W. (2000). The information bottleneck method. arXiv preprint.

---

> ### Author Response · Authors · 2025-11-27
> **Response to Reviewer Uvkf (2/2)**
>
> **W3: The core in-house dataset (e.g., 170M+ Chinese pairs, synthetic triplets) is not public, limiting reproducibility.**
>
> Thanks for raising this concern. To achieve competitive results against commercial black-box models and state-of-the-art proprietary systems, utilizing a large-scale in-house dataset was necessary, and common in practice.
> we are currently undergoing the internal approval process to open-source the curated dataset and the synthetic data generation pipeline, ensuring future reproducibility.
>
> **W4: The connector module struggles with aligning large (10B) frozen diffusion models, forcing full fine-tuning — potentially undermining modularity claims.**
>
> While we fine-tune the diffusion model in Stage 2, we argue that our approach fundamentally differs from monolithic training paradigms and remains highly economical. Crucially, we define **modularity** by the distinct functional roles of the components rather than strictly frozen parameters: the VLM remains the sole "reasoned planner," while the Diffusion Model acts strictly as the "visual synthesizer." The fine-tuning process does not entangle reasoning logic into the diffusion weights; rather, it merely adapts the diffusion features to interpret the rich "Kontext" signals. This modular design translates into significant **efficiency** gains. As illustrated in the Discussion and Figure 4, our alignment strategy requires only ~10% of the computational resources needed to train a large-scale diffusion model from scratch or an integrated multimodal transformer (e.g., BAGEL). Furthermore, regarding **convergence**, Figure 4 demonstrates that initializing with our pre-aligned VLM (via Query-Kontext) leads to significantly faster alignment compared to standard LLM-based training.
>
> **W5: Mostly quantitative; limited qualitative ablation or human preference studies.**
>
> We have added additional qualitative comparisons in the revised manuscript (**Figure 11 and 12** in Appendix). These examples consistently show that Query-Kontext achieves stronger instruction following and improved detail fidelity across diverse scenarios. In addition, we updated **Figure 1** and included further visual examples to more comprehensively illustrate model behavior.
>
> **W6: There are format issues in the cited references.**
>
> We apologize for the oversight. We have thoroughly reviewed the bibliography and corrected all formatting issues in the revised manuscript.
> Specifically, we converted entry types from @misc to @article (e.g., janusflow, qwenimage, hunyuandit), and removed duplicate citations (e.g., gpt2, llama2).
>
> Best regards,
>
> Authors of Paper 2376

---

### Official Review · Reviewer_Nkxi · 2025-11-01

**Soundness:** 3
**Presentation:** 3
**Contribution:** 2
**Rating:** 4
**Confidence:** 4

**Summary:**

1. Well-presented and easy to read; the overall format and organization are clean and consistent.
2. Tackles the unification of image understanding and image generation via a decoupled MLLM-driven conditioning interface plus a diffusion-based renderer.
3. Provides a clear dataset construction pipeline and reports concrete experimental conclusions.

**Strengths:**

1. Dataset construction, task curation, and staged training/alignment procedure are described clearly enough for reproduction.
2. Polished writing and typography; the method and training recipe are easy to follow.
3. Broad experimental coverage across T2I, editing, personalization, and multi-subject settings with sensible metrics and ablations.

**Weaknesses:**

1. Insufficient comparison with more recent unified models; subject-driven generation is benchmarked mainly against OmniGen, leaving uncertain where it stands versus newer UMM baselines.
2. The distinction from prior works that use an MLLM (or Q-Former–style bridges) as a multimodal instruction encoder is not articulated clearly.
3. Limited qualitative visualizations compared with other baselines.

**Questions:**

1. I can’t clearly tell how Query-Kontext differs from Q-Former. What advantages does Query-Kontext offer?
2. Overall, the design essentially replaces the LLM with an MLLM and appends a projector, with little architectural novelty. Moreover, although the paper claims a unified model, the evaluation on subject-driven generation compares only against OmniGen, lacking comparisons with more recent unified models.
3. Why is the text embedding concatenated again before being fed into the connector?
4. How exactly are the low-level features and the noise concatenated—what is the ordering?

---

> ### Author Response · Authors · 2025-11-27
> **Response to Reviewer Nkxi (1/2)**
>
> Dear Reviewer Nkxi:
>
> **W1: Insufficient comparison with more recent unified models; subject-driven generation is benchmarked mainly against OmniGen, leaving uncertain where it stands versus newer UMM baselines.**
>
> Thank you for raising this point. We have added comparisons with more recent unified models that report subject-driven generation results in **Tables 1** and **2**, and we also include additional qualitative comparisons in **Figure 11 and 12** (in Appendix). Query-Kontext attains the strongest DINO and CLIP-I scores on both benchmarks, together with competitive CLIP-T performance. It is worth noting that the higher CLIP-T score of OmniGen2 comes at the cost of reduced image fidelity, as illustrated in **Figure 11 and 12**.
>
> **Table 1: DreamBooth (single-subject) comparison on subject-driven generation.**
>
> | Method | DINO ↑ | CLIP-I ↑ | CLIP-T ↑ |
> | :--- | :--- | :--- | :--- |
> | Query-Kontext (ours) | **0.786** | **0.858** | 0.307 |
> | BAGEL [1] | 0.777 | 0.851 | 0.307 |
> | OmniGen2 [2] | 0.749 | 0.830 | **0.310** |
>
> **Table 2: DreamBench (multi-subject) comparison on subject-driven generation.**
>
> | Method | DINO ↑ | CLIP-I ↑ | CLIP-T ↑ |
> | :--- | :--- | :--- | :--- |
> | Query-Kontext (ours) | **0.532** | **0.731** | **0.336** |
> | BAGEL [1] | 0.439 | 0.683 | 0.335 |
> | OmniGen2 [2] | 0.488 | 0.716 | 0.332 |
>
>
> **W2: The distinction from prior works that use an MLLM (or Q-Former–style bridges) as a multimodal instruction encoder is not articulated clearly.**
>
> **Q1: I can’t clearly tell how Query-Kontext differs from Q-Former. What advantages does Query-Kontext offer?**
>
> We appreciate this suggestion and clarify the differences below:
>
> **1. Role of the queries: Decoupling vs. Compressing.**
> The core difference lies in how the architectures handle the three essential signals required for generation: **(i) high-level semantic instructions**, **(ii) coarse spatial/visual planning**, and **(iii) low-level reference details**.
>
> On the one hand, **Q-Former–style architectures** attempt to compress all three signals **((i)+(ii)+(iii))** into a single set of query tokens. This "mixed and compressed" representation often leads to an information bottleneck, causing spatial–semantic ambiguity as the queries struggle to balance semantic understanding with visual reconstruction.
> On the other hand, **Query-Kontext** adopts a **decoupled** design. We assign specific roles to different components: the *Kontext* tokens ($Q$) focus on encoding **(ii) coarse spatial/visual planning**, acting as a bridge for reasoning; the text tokens preserve **(i) explicit semantic instructions**; and the low-level encoder handles **(iii) fine-grained reference details**.
>
> Instead of merging distinct signals into a single latent channel, our design separates textual semantics, query-based contextual cues, and low-level visual features. This disentanglement allows the VLM and diffusion model to specialize in their complementary roles (reasoning vs. rendering).
>
> **2. Benefits on Training Dynamics.**
> We empirically compared three settings (as shown in Table 3): (a) our full Query-Kontext (text + queries), (b) MLP over text tokens only, and (c) Q-Former–style queries only. With the same diffusion backbone, setting (c) is significantly harder to optimize and converges to worse GenEval performance (e.g., 0.751 vs. 0.882 at 100k steps), while our decoupled design delivers both faster convergence and better final scores.
>
> Together, these aspects make Query-Kontext more than a direct Q-Former adaptation. By explicitly activating the complementary capabilities of VLMs and diffusion models through this decoupled design, our approach provides a clear architectural distinction specifically tailored for controllable and stable diffusion conditioning, distinguishing it from both Q-Former–style bridges and existing UMMs.
>
> **Table 3: Comparison of different conditioning designs during Stage 1. Results are reported using the overall score (↑) on the Geneval benchmark.**
>
> | Setting | iters | 20k | 40k | 60k | 80k | 100k |
> | :--- | :--- | :--- | :--- | :--- | :--- | :--- |
> | **a. Query-Kontext** | Text (512) & query (128) | **0.661** | **0.823** | **0.859** | **0.874** | **0.882** |
> | **b. MLP** | only Text (512) | 0.587 | 0.67176 | 0.757 | 0.770 | 0.814 |
> | **c. Qformer** | only query (640) | 0.359 | 0.596 | 0.689 | 0.732 | 0.751 |
>
> Together, these aspects make Query-Kontext more than a direct Q-Former adaptation, and specifically tailored for controllable diffusion conditioning.

---

> > ### Author Response · Authors · 2025-11-27
> > **Response to Reviewer Nkxi (2/2)**
> >
> > **W3: Limited qualitative visualizations compared with other baselines.**
> >
> > Thank you for this suggestion. We have added more qualitative comparisons in **Figure 11 and 12** (in Appendix), which clearly show that Query-Kontext better preserves identity (e.g., fine-grained shapes and textures of reference objects) while still following textual instructions. In addition, we updated **Figure 1** and included further visual examples to more comprehensively illustrate model behavior.
> >
> > **Q2: Overall, the design essentially replaces the LLM with an MLLM and appends a projector, with little architectural novelty. Moreover, although the paper claims a unified model, the evaluation on subject-driven generation compares only against OmniGen, lacking comparisons with more recent unified models.**
> >
> > As discussed above, our architectural contribution lies in introducing *Kontext tokens*, which explicitly activate and separate the complementary capabilities of the VLM (multimodal reasoning) and the diffusion model (high-fidelity rendering). This mechanism has not appeared in prior UMMs, which generally merge heterogeneous signals into a single latent channel.
> >
> > Furthermore, we emphasize our **novel training paradigm**. Query-Kontext employs a unique three-stage progressive curriculum that systematically aligns the MLLM with diffusion decoders of increasing capacity. This decoupled strategy is not merely an implementation detail but a core design choice that prevents capacity competition between reasoning and rendering modules. This ensures economical alignment without catastrophic forgetting, which we found critical for achieving high performance on demanding editing and customization tasks.
> >
> > Regarding comparisons, we note that many recent UMMs lack the explicit control mechanisms required to evaluate the full spectrum of downstream tasks, particularly those demanding strict pixel-level fidelity (e.g., instruction editing) and identity preservation (e.g., single/multi-subject customization). Our shifted-RoPE design addresses this limitation, enabling unified performance where others struggle. To address your concern, we have included additional quantitative baselines in Tables 1 and 2. Additionally, we have added more qualitative comparisons in **Figure 11 and 12**.
> >
> > **Q3: Why is the text embedding concatenated again before being fed into the connector?**
> >
> > The concatenation serves as a Learning-efficiency trade-off. Including text embeddings preserves structured semantic information and reduces the learning burden on query tokens. Increasing the number of learnable queries and removing text tokens, as in Table 3(c), leads to optimization instability similar to Q-Former–style architectures, which is why we do not adopt that design.
> >
> > As shown in Table 3(b), using only MLLM text tokens yields final performance close to the original in-house MMDiT (0.814 vs. 0.819), but still below Query-Kontext (Table 3). The performance gap becomes even larger in downstream image editing and reference-to-image tasks, where semantic precision is vital.
> >
> > **Q4: How exactly are low-level features and noise concatenated? What is the ordering?**
> >
> > We apologize for the earlier ambiguity. As shown in the left subfigure of Figure 2, we adopt two concatenation orders depending on task requirements: 1) for the tasks requiring strict pixel fidelity (e.g., instruction-based editing), we `concat(low-level features, noise)` with the low-level features assigned a shifted RoPE in the negative spatial direction.
> > 2) for the asks emphasizing identity preservation with flexibility (e.g., customized generation), we `concat(noise, low-level features)` with low-level features shifted along the positive direction. We have updated the clarified description in the revised manuscript (Sec 3.1).
> > This shifted RoPE mechanism is essential for distinguishing source images from reference images and for enabling unified handling of diverse tasks within a single model.
> >
> > We hope these clarifications and new results address your concerns and better highlight the novelty and practical value of Query-Kontext as a unified multimodal model for image generation and editing.
> >
> >
> > Best regards,
> >
> > Authors of Paper 2376

---

> ### Comment · Reviewer_Nkxi · 2025-11-28
>
> The rebuttal addresses most of my concerns, and several clarifications are helpful. However, a few key additions should be incorporated into the revised PDF to make the improvements verifiable and easy to follow.
>
> I still confuse about Q3. The rebuttal offers a motivation, but the explanation is still somewhat high-level. Please add a clearer implementation-level rationale in the revised PDF (what exactly is concatenated, at which interface, and why it is necessary vs. text-only or query-only), ideally supported by a small ablation.

---

> ### Author Response · Authors · 2025-11-29
> **Official Comment by Authors**
>
> We sincerely thank the reviewer for their prompt feedback. As suggested, we have further incorporated the following updates, **highlighted in blue** in the newest revised manuscript:
>
> 1. We have updated **Table 6** and **Table 7** in the main text to include additional comparative results against state-of-the-art UMMs.
> 2. We have added a clearer description in **Section 3.1** regarding the interaction mechanism between *kontext* tokens, text latents, and low-level features at the implementation level.
> 3. We have merged the aforementioned ablation study (in **Appendix C.5 **) into the response to **W2**. This experiment is critical for demonstrating that ``kontext'' tokens serve as a necessary interface compared to traditional Q-Former and MLP bridging approaches.
>
> We hope these clarifications have resolved your concerns. Thank you once again for your thoughtful review of our submission.

---

### Author Response · Authors · 2025-11-27
**General response (1/2)**

We thank the reviewers for their insightful feedback and are encouraged that they unanimously recognized the core value of our work. Specifically, reviewers commended the **elegant architectural decoupling** achieved by our "Kontext" tokens, which effectively separates multimodal reasoning from high-fidelity rendering. The **strong empirical performance** was also highlighted, with reviewers noting our framework's ability to match or outperform state-of-the-art baselines across diverse unified tasks like instruction editing and multi-subject composition. Additionally, the extensive dataset curation and the clarity of our **three-stage progressive training curriculum** were appreciated for facilitating reproducibility and providing a solid foundation for industrial-grade unified models.

We have made the following updates to the paper, highlighted in **blue** in the revised manuscript:

*   **Update:** Expanded **Figure 1** with additional qualitative examples to more comprehensively illustrate model behavior across different tasks, as suggested by Reviewer Nkxi.
*   **Clarification:** Added concatenation ordering details in Section 3.1, specifying strategies for pixel fidelity vs. identity preservation, as asked by **Reviewer Nkxi**.
*   **Correction:** Adjusted the position of the caption for **Table 2** to appear above the table, as pointed out by Reviewer fgrd.
*   **Addition:** Included extensive qualitative comparisons against recent unified models (BAGEL, OmniGen2) on DreamBooth and DreamBench benchmarks in **Appendix C.1 (Figure 11 and Figure 12)**, addressing the requests for visual comparisons from Reviewers Nkxi and Uvkf.
*   **Addition:** Provided visualizations comparing Stage 1 and Stage 2 outputs to empirically validate the decoupling of reasoning (VLM) and synthesis (Diffusion) in **Appendix C.2 (Figure 13)**, as requested by Reviewer Uvkf.
*   **Addition:** Added qualitative results demonstrating generalization capabilities on unseen concepts (e.g., specific brands, artistic styles) and zero-shot outpainting tasks in **Appendix C.3 (Figure 14)**, addressing Reviewer fgrd's concern regarding compositional reasoning.
*   **Addition:** Included a visual ablation study for the low-level encoder to demonstrate its specific contribution to fine-grained editing details in **Appendix C.4 (Figure 15)**, as requested by Reviewer Asae.
*   **Correction:** Corrected all formatting issues in the revised manuscript, as requested by Reviewer Uvkf.


Furthermore, we sincerely invite the reviewers to examine the Appendix for additional qualitative results (Figures 11-15), the detailed response regarding theoretical justification as asked by Reviewer Uvkf (W1), and the clarification on training and architectural novelty as asked by Reviewer Nkxi (W2). We warmly encourage you to review these updates in the revised manuscript and the detailed responses below. We hope our response and additional experiments address your concerns!


Best regards,

Authors of Paper 2376

---

> ### Author Response · Authors · 2025-12-03
> **General response (2/2)**
>
> Based on the discussion with reviewers, we have further incorporated the following updates, **highlighted in blue** in the newest revised manuscript:
> *   **Clarification:** Added a clearer description in **Section 3.1** regarding the interaction mechanism between *kontext* tokens, text latents, and low-level features at the implementation level.
> *   **Addition:** Included an ablation study in **Appendix C.5** demonstrating that ``kontext'' tokens serve as a necessary interface compared to traditional Q-Former and MLP bridging approaches.
> *   **Update:** Updated **Table 6** and **Table 7** in the main text to include additional comparative results against state-of-the-art UMMs.

---

### Author Response · Authors · 2025-12-03
**Review and Reviewer-Author Discussion Summary**

Dear PCs, SACs, ACs, and Reviewers,

We sincerely thank the reviewers for their constructive feedback and the engaging discussion period. To assist the Area Chair in their decision-making, we provide a summary of the key strengths recognized by the reviewers and how we addressed their concerns.

**Strength.** We are encouraged that the reviewers unanimously recognized the core value of our work, specifically:

1. **Elegant Architectural Decoupling:** All four reviewers commended the \"Kontext\" token design, which effectively separates multimodal reasoning (VLM) from high-fidelity rendering (Diffusion) (Uvkf: \"explicitly separates reasoning... from rendering\", fgrd: \"Elegant decoupling\", Asae: \"innovatively bridges VLMs and diffusion\").

2. **Strong Empirical Performance:** Reviewers highlighted our framework's ability to match or outperform state-of-the-art baselines across diverse tasks (Nkxi: \"Broad experimental coverage\", fgrd: \"Strong empirical performance... outperforming several unified... baselines\", Asae: \"matches strong unified baselines\").

3. **Comprehensive Dataset and Curriculum:** The extensive dataset curation and the clarity of the three-stage progressive training strategy were praised for facilitating reproducibility (Nkxi: \"Dataset construction... described clearly\", Uvkf: \"Three-stage progressive training gradually enforces this separation\", fgrd: \"Extensive dataset curation\").

**Concerns and Our Addressing.** During the discussion period, we actively addressed the reviewers' concerns with new experiments and clarifications, leading to positive acknowledgments:

*   **Comparisons with Recent Unified Models (Nkxi, fgrd):** Reviewers requested comparisons against newer baselines like OmniGen2 and evaluations on unseen concepts.

*   *Our Addressing:* We added quantitative comparisons (Tables 6, 7  in the revision) and qualitative visualizations (Figures 11, 12  in the revision)  showing Query-Kontext outperforms current UMMs. We also added experiments on unseen-concept editing and zero-shot outpainting (Figure 14 in the revision), and update the Figure 1 in the revision to include further visual examples . Reviewer Nkxi acknowledged these updates addressed their concerns.

*   **Theoretical and Architectural Justifications (Uvkf, Nkxi, Asae):** Reviewers asked for the theoretical basis of the decoupling strategy and the distinction from Q-Former architectures.

*   *Our Addressing:* We articulated the Information Bottleneck principle as the theoretical grounding for our design and provided comparisons (Table 9, Figure 13 in the revision) demonstrating that our decoupled approach converges faster and performs better than Q-Former-style bridges. Reviewers Uvkf and Nkxi found these clarifications helpful.

*   **Ablation of Key Mechanisms (Uvkf, Asae):** Reviewers requested validation for the Shifted-RoPE and the Stage 3 low-level encoder.

*   *Our Addressing:* We provided ablation studies (Tables 4, 5 in the response) confirming that Shifted-RoPE is essential for multi-subject generation, and added visualizations (Figure 15 in the revision) showing the low-level encoder's contribution to fine-grained editing. Reviewers Uvkf and Asae accepted these results.

*   **Data Reproducibility and Fairness (Uvkf, Asae, fgrd):** Reviewers noted the use of in-house data.

*   *Our Addressing:* We clarified our internal approval process to open-source the dataset and pipeline to ensure reproducibility. While Reviewer Asae retained a note on data fairness, other reviewers expressed satisfaction with our commitment to open-sourcing.

We believe we have addressed the majority of concerns, evidenced by the positive feedback and the maintenance of positive scores. Specifically, Reviewer **Asae** stated, \"Thank you, the clarifications on Q1 and Q2 are helpful, and I no longer have any concerns about them,\" and Reviewer **Nkxi** acknowledged, \"The rebuttal addresses most of my concerns, and several clarifications are helpful.\" Additionally, Reviewer **fgrd** affirmed, \"I will keep my positive score.\" Although Reviewer **Uvkf** was unable to provide further feedback due to rule changes, we remain confident that our detailed response covers their points. We thank the reviewers and ACs for their time and effort.

Sincerely,

Authors

---

### Meta-Review · Area_Chair_FdNr · 2026-01-05

**Summary:**

The reviewers concerns' focus on 1) technical novelty, 2) reproducibility and fairness, 3) theoretical justification and ablation, and 4) insufficient comparison.

1. Reviewers Nkxi points out that the difference to Q-Former is not clearly discussed, and the replacement of an LLM with an MLLM possesses limited novelty.

2. Reviewers Uvkf and Asae point out that the model depends heavily on an in-house dataset, which is skewed towards Chinese. This could lead to compromised reproducibility and fairness.

3. Reviewer Uvkf argue that the paper lacks theoretical explanation why the decoupling of reasoning and synthesis is effectively. Reviewer Asae requests more ablations of Shifted-ROPE, fn find that the definition of "kontext" tokens unclear.

4. Reviewer Nkxi mentions that there is insufficient comparison with recent unified models. Reviewer fgrd notes that this paper lack targeted evaluation on "compositional reasoning" or "unseen-concept editing".

**Reviewer Concerns:**

Many of the concerns, including comparison to baselines, ablations of Shifted-ROPE, evaluation of unseen concepts, theoretical justification, are addressed. However, the concerns regarding data fairness and implementation rationale for concatenation remain unaddressed.

**Reviewer Scores:**

This paper receives ratings of (4, 4, 6, 6), and the AC anticipates increased ratings of (4, 5, 6, 6) after discussion. Given the marginal ratings, the AC agrees with the concerns of the reviewers. The authors are advised to address the concerns of the reviewers. A reject is recommended.

---

### Decision · Program_Chairs · 2026-01-26

Reject